# Training-Free Hashing-Based Attention via Binary Principal Components

**Daohai Yu** [1]   **Zhanpeng Zeng** [1]   **Keyu Chen** [2]   **Wenhao Li** [1]   **Zhifeng Shen** [2]
**Luxi Lin** [1]   **Ruizhi Qiao** [2]   **Xing Sun** [2]   **Rongrong Ji** [1][3]

## Abstract

Long-context large language models (LLMs) are increasingly deployed in real-world applications, yet self-attention remains a major efficiency bottleneck – especially during decoding – due to the necessity of repeatedly processing ever-growing key-value (KV) caches. Existing sparse attention reduce computation by attending to fewer KV pairs, but often suffer from substantial accuracy degradation, require additional training, or rely on expensive hashing. In this work, we present **BinaryPC**, a training-free, data-aware hashing-based sparse attention for long-context LLMs. BinaryPC constructs compact binary hash codes and corresponding hash function by computing binary principal components of data. Unlike Locality-Sensitive Hashing (LSH) with data-independent random projections or learned non-linear hashing methods, BinaryPC constructs binary codes that explicitly preserve the structural information of data without requiring gradient-based training. Comprehensive experiments across multiple model families and long-context benchmarks show that BinaryPC preserves accuracy relative to full attention while achieving superior performance among sparse and hashing-based baselines. On modern GPUs, BinaryPC improves end-to-end decoding throughput by $3.56\times$ over the FlashAttention kernel. Our code is available at https://github.com/yudaohai666/BPC.

## 1. Introduction

With the rapid advancement of large language models (LLMs), handling long-range dependencies has become

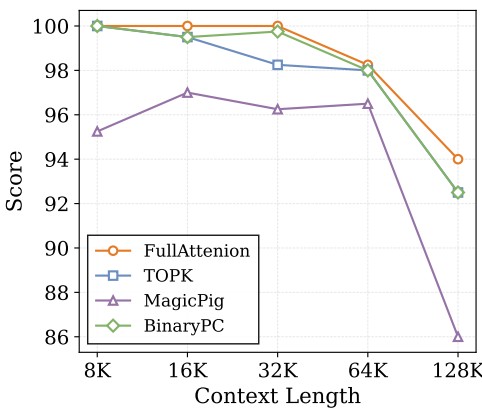

*Figure 1.* Performance comparison on the RULER (Hsieh et al., 2024) NIAH multi-value task. Oracle TOPK selects top-$k$ keys using exact full-precision attention with 2% budget.

essential for applications such as multi-document question answering (Wang et al., 2024), conversational agents, and complex reasoning tasks (Achiam et al., 2023; Anthropic, 2024; Yang et al., 2025a). Modern LLM inference is typically divided into two stages: the prefill stage, where input tokens are processed in parallel to construct the key-value (KV) cache (Pope et al., 2023), and the decoding stage, where tokens are generated autoregressively. Compared to the highly parallel prefill stage, decoding involves frequent memory transfers for the growing KV cache while per-token generation remains inherently sequential. This results in low GPU utilization and suboptimal hardware efficiency, severely limiting throughput for long sequences (He & Zhai, 2024).

To mitigate this issue, prior research has explored a range of sparse attention mechanisms that reduce computational overhead during decoding. Static selection strategies (Ge et al., 2024; Li et al., 2024; Cai et al., 2025; Qin et al., 2025; Lin et al., 2025) typically aggregate or prune KV entries after prefilling, enabling subsequent decoding to operate on a compressed representation. Alternative methods maintain a fixed memory budget during decoding by continuously discarding or down-weighting less critical tokens (Zhang et al., 2023; Oren et al., 2024; Xiao et al., 2024; Adnan et al., 2024). Additionally, the query-aware selection (Tang et al., 2024) dynamically identifies salient token subsets at each decoding step, using heuristics derived from the current

[1]Key Laboratory of Multimedia Trusted Perception and Efficient Computing, Ministry of Education of China, Xiamen University, 361005, P.R. China. [2]Tencent YouTu Lab, Shenzhen, China [3]Sino-Russian Research Center for Digital Economy. Correspondence to: Zhanpeng Zeng <zzeng@xmu.edu.cn>.

*Proceedings of the $43^{rd}$ International Conference on Machine Learning*, Seoul, South Korea. PMLR 306, 2026. Copyright 2026 by the author(s).

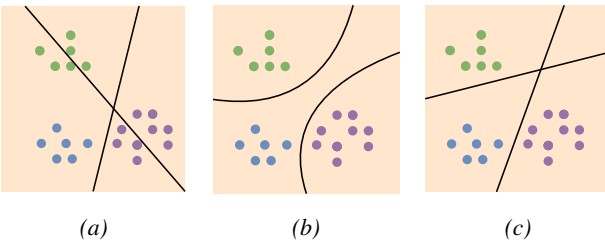

*Figure 2.* **Illustration of different hashing methods.** (a) Locality-Sensitive Hashing with random projections (Chen et al., 2025); (b) Learned hashing with trained mappings (Li et al., 2025); (c) BinaryPC with binary principal directions.

query. However, these techniques often incur performance degradation, as query-agnostic eviction can remove relevant evidence, while heuristic saliency estimates may fail to align with true attention affinity.

Recently, hashing-based sparse attention, such as MagicPIG (Chen et al., 2025) and Spotlight (Li et al., 2025), has emerged as a promising paradigm for fine-grained salient token selection. By leveraging compact binary representations, these methods enable efficient similarity computation by significantly reducing memory transfer and enabling low cost bitwise operations. However, MagicPIG relies on data-independent projections, requiring extremely long hash codes to compensate for precision loss and may still limit its ability to accurately identify true attention affinity. As illustrated in Figure 1, on the NIAH multi-value task, MagicPIG consistently underperforms oracle TOPK across all context lengths. At a 128K context length, MagicPIG falls 6.5 percentage points below oracle TOPK, which itself remains nearly identical to the full-attention baseline. This performance gap highlights the limitations of data-agnostic projections, which fail to capture the structural properties inherent in LLM activations. As illustrated in Figure 2a, data-independent random hyperplanes that are oblivious to the underlying data structure lead to suboptimal hash partitions. In contrast, learned non-linear hashing functions such as Spotlight (Li et al., 2025) can better capture the structure of the data (Figure 2b), but incur substantial training overhead, requiring calibration data and extensive optimization for each model variant. These observations motivate the need for a training-free yet data-aware hashing scheme.

To address these limitations, we propose **BinaryPC**, a training-free data-aware hashing-based sparse attention. Given key vectors $\mathbf{K}$, BinaryPC constructs compact binary hash codes $\mathbf{H}$ and a hashing projection $\mathbf{P}$ by computing binary principal components of $\mathbf{K}$, which minimizes the reconstruction error $\|\mathbf{K} - \mathbf{HP}\|_F$. In this formulation, the binary hash codes $\mathbf{H}$ capture the structural information of $\mathbf{K}$ through the projection $\mathbf{P}$. Unlike LSH with data-independent random projections or learned hashing methods requiring per-model optimization, BinaryPC derives hash codes and projections in a single forward pass

without gradient-based training. As shown in Figure 2c, BinaryPC leverages the principal directions of the data to achieve structural alignment.

Notably, BinaryPC achieves competitive retrieval fidelity with compact 64-bit codes—over $10\times$ shorter than MagicPIG (Chen et al., 2025) and $2\times$ shorter than Spotlight (Li et al., 2025)—substantially reducing memory transfer and computational overhead. Furthermore, the reconstruction-based formulation provides interpretable error signals, enabling an error-aware safeguard that preserves recall for hard-to-hash tokens. BinaryPC is model-agnostic and can be integrated seamlessly into pre-trained LLMs via online projection during inference or offline calibration, yielding up to $3.56\times$ improvement in end-to-end decoding throughput over FlashAttention (Dao et al., 2022; Dao, 2024) and $5.04\times$ speedup when FlashAttention falls back to standard implementations.

Our contributions are summarized as follows:

- We propose **BinaryPC**, a training-free data-aware hashing-based sparse attention mechanism that constructs compact binary codes by computing binary principal components of data, enabling efficient retrieval via fast bitwise operations.

- We develop a lightweight procedure that minimizes the reconstruction error $\|\mathbf{K} - \mathbf{HP}\|_F$, allowing the binary hash codes $\mathbf{H}$ to capture the structural information of $\mathbf{K}$ through the projection $\mathbf{P}$. As a result, salient token retrieval can be performed both efficiently and accurately using $\mathbf{H}$. An error-aware safeguard is incorperated to preserve recall for hard-to-hash tokens.

- Extensive experiments show that BinaryPC maintains accuracy across multiple models and benchmarks while substantially improving decoding throughput.

## 2. Related Works

### 2.1. Sparse Attention

To mitigate the computational bottlenecks of long-context LLMs, sparse attention mechanisms selectively reduce the number of KV pairs during attention calculation.

A primary category relies on static selection strategies determined during the prefill stage. StreamingLLM (Xiao et al., 2024) retains attention sinks with a sliding window; FastGen (Ge et al., 2024) adaptively selects policies based on prefill attention patterns; SnapKV (Li et al., 2024) identifies significant clusters within an observation window to retain salient KV representations; PyramidKV (Cai et al., 2025) implements layer-wise pyramidal budget allocation; CAKE (Qin et al., 2025) and CompressKV (Lin et al., 2025) leverage attention entropy and semantic retrieval heads for

fine-grained optimization. However, these methods are query-agnostic: token importance is assessed only once after prefilling, leading to irreversible information loss when subsequent queries depend on unprioritized context.

Dynamic selection methods perform token filtering at each decoding step for query-awareness. H2O (Zhang et al., 2023) tracks heavy-hitter tokens via cumulative attention scores; TOVA (Oren et al., 2024) approximates query-key inner products for online selection; Keyformer (Adnan et al., 2024) utilizes regularized importance estimates for stability. Quest (Tang et al., 2024) partitions tokens into blocks for page-level selection based on approximated maximum attention scores. These approaches often rely on coarse-grained or heuristic scoring that may misalign with true attention affinity. Moreover, block-level strategies suffer from intra-page fragmentation, requiring entire pages to be retained even when only a small subset of tokens is salient.

### 2.2. Hashing-Based Sparse Attention

By leveraging compact binary representations, hashing-based sparse attention enables efficient token-level retrieval via bitwise operations, facilitating finer-grained token selection than coarse-grained heuristic strategies.

MagicPIG (Chen et al., 2025) employs Locality-Sensitive Hashing (LSH) with data-independent random projections to select salient tokens. However, LSH's random hyperplanes are often misaligned with the intrinsic structure observed in LLM activations. This misalignment necessitates extremely long hash codes (exceeding 1000 bits) to maintain acceptable recall, and further requires auxiliary structures to stabilize retrieval, increasing system complexity and memory overhead. Learned hashing methods (Desai et al., 2025; Gong et al., 2025; Li et al., 2025) incorporate trainable hashing functions to more effectively capture query-key similarity. Spotlight (Li et al., 2025) replaces the data-independent LSH with a learned MLP-based hashing function trained with ranking-oriented objectives. By fitting the underlying data distribution, Spotlight reduces the hash length while preserving the retrieval accuracy. However, it introduces considerable training overhead. It requires around 8 hours of optimization on 8,192 samples even for 8K context length. Moreover, the hashing module must be retrained for each model, restricting its practicality as a drop-in replacement.

In contrast to MagicPIG that require long codes and Spotlight that demands extensive training, BinaryPC achieves superior performance with compact 64-bit codes. This is accomplished through a lightweight training-free data-aware procedure that explicitly minimizes the reconstruction error $\|\mathbf{K} - \mathbf{HP}\|_F$, which in turn allows salient tokens to be identified both efficiently and accurately.

## 3. BinaryPC

### 3.1. Hashing-Based Retrieval

Efficiently and accurately identifying salient tokens is fundamental to sparse attention mechanisms. Hashing-based methods approximate attention scores with substantially reduced computational cost to efficiently retrieve the top-$k$ most relevant key-value (KV) pairs.

Given query $\mathbf{q}$ and key $\mathbf{k}$, existing methods typically transform these vectors into compact binary codes $\mathbf{h} \in \{-1, 1\}^H$ via a hashing function $\Phi(\cdot)$, yielding $\mathbf{h}_q = \Phi(\mathbf{q})$ and $\mathbf{h}_k = \Phi(\mathbf{k})$, where $H < D$. For instance, MagicPIG (Chen et al., 2025) employs LSH with random projections, whereas Spotlight (Li et al., 2025) adopts a learnable MLP-based hashing scheme to adapt to data distributions.

The objective of these methods is to ensure that the inner product between $\mathbf{h}_q$ and $\mathbf{h}_k$ serves as a high-fidelity proxy for the full precision query-key inner product:

$$\mathbf{h}_q \mathbf{h}_k^\top \sim \mathbf{q} \mathbf{k}^\top, \tag{1}$$

When this relationship holds, relevant KV pairs can be identified via $\mathbf{h}_q \mathbf{h}_k^\top$, which can be calculated with significantly reduced memory transfer and lower computational cost.

### 3.2. BinaryPC

Obtaining a high-fidelity proxy for the full-precision query–key inner product in practice is challenging, as existing approaches either rely on data-independent projections that poorly align with model activations or require costly training to adapt to specific models.

To address this gap, we develop BinaryPC, a training-free and data-aware hashing-based attention mechanism. Rather than applying data-aware hashing symmetrically to both query and keys, BinaryPC adopts an asymmetric design that processes queries and keys differently. Since the memory transfer of keys dominates the cost due to the large number of KV pairs relative to a single query, we focus first on efficiently encoding keys into binary representations. Specifically, BinaryPC derives binary hash codes for keys directly from the geometry of the key vectors, constructing compact binary hash codes that preserve the structural information of the key vectors. We will discuss query encoding and the complete retrieval procedure later in this section.

**Problem Formulation.** Given $N$ key vectors $\mathbf{K} \in \mathbb{R}^{N \times D}$, BinaryPC computes binary hash codes $\mathbf{H} \in \{-1, 1\}^{N \times H}$ and a real-valued projection matrix $\mathbf{P} \in \mathbb{R}^{H \times D}$ that minimizes the reconstruction error:

$$\min_{\mathbf{H}, \mathbf{P}} \|\mathbf{K} - \mathbf{HP}\|_F. \tag{2}$$

Once optimal $\mathbf{H}$ and $\mathbf{P}$ are found, for any key $\mathbf{k}$ in $\mathbf{K}$ with corresponding hashcode $\mathbf{h}$ in $\mathbf{H}$, since $\mathbf{k} \approx \mathbf{hP}$, for any

query $\mathbf{q}$, a high-fidelity proxy can be achieved via

$$(\mathbf{q}\mathbf{P}^\top)\mathbf{h}^\top = \mathbf{q}(\mathbf{h}\mathbf{P})^\top \approx \mathbf{q}\mathbf{k}^\top. \qquad (3)$$

Note that $\mathbf{q}\mathbf{P}^\top$ is real-valued. Accordingly, the second component of this asymmetric design maps $\mathbf{q}\mathbf{P}^\top$ into a binary representation through quantization and bit arrangement, enabling fully bitwise operations for computational efficiency. We will discuss this procedure later.

**Binary Principal Components Finding.** Note that Eq. (2) is closely related to Principal Component Analysis (PCA). PCA can be solved via iteratively finding the principal component (or the largest singular vector) and removing the component from the signal. Inspired by PCA, we solve Eq. (2) by iteratively finding binary principal component and removing this component from $\mathbf{K}$, as detailed in Algorithm 1. Note that given a randomly sampled vector $\mathbf{v}^*$, let $\mathbf{R}$ be the residual signal (initially $\mathbf{R} = \mathbf{K}$), the principal component of $\mathbf{R}$ can be discovered via

$$\mathbf{u} = (\mathbf{R}\mathbf{R}^\top)^n \mathbf{R}\mathbf{v}^{*\top} \qquad (4)$$

for sufficiently large $n$, since the singular values of $\mathbf{R}$ will grow exponentially fast as $n$ increases making the largest singular vector dominate. However, since we are interested in the binary quantized component, exact principal component is less useful. We use

$$\mathbf{u} = \text{sign}(\mathbf{R}\mathbf{v}^{*\top}) \qquad (5)$$

as a sufficiently good binary component to save computation since the largest singular value ensure $\mathbf{R}\mathbf{v}^{*\top}$ will lean towards the principal component with high probability. Then,

$$\mathbf{v} = \mathbf{u}\mathbf{R}/N \qquad (6)$$

finds the magnitudes of projecting columns of $\mathbf{R}$ onto $\mathbf{u}$. Lastly, the component $\mathbf{u}$ is removed from columns of $\mathbf{R}$ via $\mathbf{R} - \mathbf{u}^\top\mathbf{v}$. Through this iterative procedure, we can construct both hash codes $\mathbf{H}$ and projection $\mathbf{P}$. Appendix A.3 provides a visual overview of this iterative rank-1 binary decomposition process. The left plot of Figure 3 shows that as the iterative procedure progress, $||\mathbf{R}||_F$ become progressively smaller, indicating the effectiveness of Algorithm 1 in solving Eq. (2).

In an alternative setting where the projection matrix $\mathbf{P}$ is given, we compute the hash code of a key vector $\mathbf{k}$ by finding a binary representation of $\mathbf{k}$ using rows of $\mathbf{P}$ as the "basis". Since the rows of $\mathbf{P}$ might not be orthogonal. This procedure should be done iteratively. Let $\mathbf{r}$ be the residual signal (initially $\mathbf{r} = \mathbf{k}$), for each row $\mathbf{v}$ of $\mathbf{P}$, we find the binarized scalar $\text{sign}(\mathbf{r}\mathbf{v}^\top)$ and subtract the corresponding component from $\mathbf{r}$. The full procedure is summarized in Algorithm 2. The right plot of Figure 3 shows that as the iterative procedure progress, $||\mathbf{r}||_2$ become progressively

---

**Algorithm 1** Constructing hash codes and corresponding hashing projection

**Input:** key vectors $\mathbf{K} \in \mathbb{R}^{N \times D}$ and target hash code length $H$
Initialize empty $\mathbf{H} \in \{-1, 1\}^{N \times H}$ and $\mathbf{P} \in \mathbb{R}^{H \times D}$
Initialize residual $\mathbf{R} \leftarrow \mathbf{K}$
**for** $i = 1$ **to** $i = H$ **do**
    sample $\mathbf{v}^* \sim N(0, \mathbf{I}_D)$
    compute $\mathbf{u} \leftarrow \text{sign}(\mathbf{R}\mathbf{v}^{*\top})$ and $\mathbf{v} \leftarrow \mathbf{u}\mathbf{R}/N$
    update $\mathbf{R} \leftarrow \mathbf{R} - \mathbf{u}^\top\mathbf{v}$
    $\mathbf{H}[:, i] \leftarrow \mathbf{u}$ and $\mathbf{P}[i, :] \leftarrow \mathbf{v}$
**end for**
**Output:** hash codes $\mathbf{H}$ and projection $\mathbf{P}$

---

**Algorithm 2** Given projection, computing hashcode

**Input:** key vector $\mathbf{k} \in \mathbb{R}^D$ and projection $\mathbf{P} \in \mathbb{R}^{H \times D}$
Initialize empty $\mathbf{h} \in \{-1, 1\}^H$
Initialize residual $\mathbf{r} \leftarrow \mathbf{k}$
**for** $i = 1$ **to** $i = H$ **do**
    let $\mathbf{v} \leftarrow \mathbf{P}[i, :]$
    compute $u \leftarrow \text{sign}(\mathbf{r}\mathbf{v}^\top)$
    update $\mathbf{r} \leftarrow \mathbf{r} - u\mathbf{v}$
    $\mathbf{h}[i] \leftarrow u$
**end for**
**Output:** hashcode $\mathbf{h}$

---

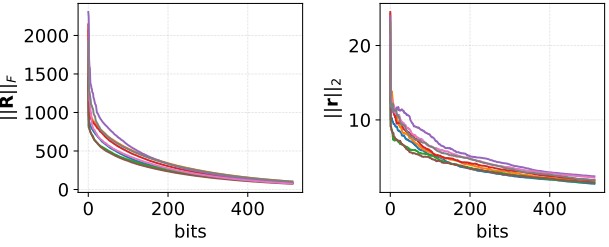

*Figure 3.* Left: the visualization of Frobenius norm of $\mathbf{R}$ vs hashcode length $H$ for Algorithm 1. Right: plot is the visualization of L2 norm of $\mathbf{r}$ vs hashcode length $H$ for Algorithm 2. Different curves represent different data. The algorithms converge in all tested data.

smaller, indicating the effectiveness of Algorithm 2 in representing $\mathbf{k}$ using rows of $\mathbf{P}$ as the "basis".

**Offline Projection Calibration (OPC).** Prior text discusses the procedure of constructing the projection $\mathbf{P}$ given $\mathbf{K}$ in an online setting. Alternatively, $\mathbf{P}$ can be computed offline using a small calibration dataset prior to deployment to save some computational overhead. To ensure diverse coverage across different textual domains, we construct a calibration set comprising 180 samples drawn from three representative corpora: PG19 (Rae et al., 2020) for long-form literary text, ProofPile (Zhangir Azerbayev, 2022) for mathematical and formal reasoning content, and CodeParrot (Zhang, 2024) for programming code, with 60 samples from each source. Then, we perform a forward pass to collect key vectors from each attention layer. Algorithm 1 is then applied to the aggregated keys to derive these projections.

*Table 1.* Performance on InfiniteBench (Zhang et al., 2024) (left) and LongBench v2 (Bai et al., 2025) (right).

| Methods | Token | InfiniteBench | | | | | | | | LongBench v2 | | | | | |
| | | En.Sum | En.QA | En.MC | En.Dia | Zh.QA | Math.F | R.PK | Avg. | Easy | Hard | Short | Medium | Long | Avg. |
|---------|-------|--------|-------|-------|--------|-------|--------|------|------|------|------|-------|--------|------|------|
| **Llama-3.1-8B** | Full | 32.13 | 25.78 | 69.00 | 19.50 | 31.90 | 25.43 | 99.32 | 43.30 | 31.2 | 28.3 | 33.3 | 27.0 | 27.8 | 29.4 |
| PyramidKV | 2K | 27.87 | 25.83 | 69.00 | 19.00 | 31.51 | 25.43 | 99.32 | 42.57 | 30.7 | 28.0 | 33.9 | 26.0 | 26.9 | 29.0 |
| Cake | 2K | 28.58 | 25.52 | 69.00 | 18.00 | 30.80 | 25.43 | 99.32 | 42.38 | 31.8 | 28.6 | 33.9 | 27.4 | 27.8 | **29.8** |
| CompressKV | 2K | 29.84 | 25.79 | 69.00 | 19.50 | 31.60 | 25.43 | 99.32 | 42.92 | 31.2 | 28.3 | 33.3 | 26.5 | 28.7 | 29.4 |
| Quest | 2K | 17.63 | 24.61 | 69.43 | 18.50 | 31.29 | 25.43 | 99.32 | 40.89 | 28.1 | 26.0 | 31.1 | 24.7 | 24.1 | 26.8 |
| MagicPIG | Default | 32.10 | 25.71 | 69.00 | 20.00 | 30.70 | 25.43 | 99.32 | 43.18 | 29.7 | 26.4 | 33.9 | 24.7 | 23.1 | 27.6 |
| BinaryPC | 2% | 33.06 | 24.82 | 69.00 | 19.00 | 31.87 | 25.43 | 99.32 | **43.21** | 32.8 | 28.0 | 35.0 | 26.0 | 28.7 | **29.8** |
| BinaryPC | 2K | 32.30 | 25.19 | 69.00 | 18.50 | 31.98 | 25.43 | 99.32 | 43.10 | 32.8 | 28.0 | 33.9 | 27.0 | 28.7 | **29.8** |
| BinaryPC w/ OPC | 2% | 32.61 | 25.44 | 69.00 | 17.50 | 32.35 | 25.43 | 99.32 | 43.09 | 31.8 | 28.6 | 33.9 | 27.4 | 28.7 | 29.4 |
| BinaryPC w/ OPC | 2K | 32.23 | 25.28 | 69.00 | 18.00 | 32.32 | 25.43 | 99.32 | 43.08 | 31.8 | 27.7 | 34.4 | 25.1 | 28.7 | 29.2 |

---

**Algorithm 3** Computing hashing score

---

**Input:** query vector $\mathbf{q} \in \mathbb{R}^D$, hashing projection $\mathbf{P} \in \mathbb{R}^{H \times D}$, and hashcode $\mathbf{h} \in \{-1, 1\}^H$
Compute quantized $\mathbf{q}\mathbf{P}^\top$, where sign bits are represented as $\mathbf{s} \in \{-1, 1\}^H$ and magnitude bits are represented as $\mathbf{M} \in \{0, 1\}^{7 \times H}$
Compute $\mathbf{s} \leftarrow \mathbf{h} \text{ XOR } \mathbf{s}$
Initialize scores $c = 0$
**for** $i = 1$ **to** $i = 7$ **do**
    $m \leftarrow$ count-bits($(\mathbf{s} \wedge \mathbf{M}[i, :])$) $-$ count-bits($\neg\mathbf{s} \wedge \mathbf{M}[i, :]$)
    Update $c \leftarrow c + m \ll i$
**end for**
**Output:** hash score $c$

---

**Retrieval Phase.** During inference, the projection $\mathbf{P}$ is applied to the incoming query, $\mathbf{q}\mathbf{P}^\top$, and a hash score will be calculated:

$$c = (\mathbf{q}\mathbf{P}^\top)\mathbf{h}^\top \tag{7}$$

As discussed, we map $\mathbf{q}\mathbf{P}^\top$ into a binary representation. Specially, the projected query $\mathbf{q}\mathbf{P}^\top \in \mathbb{R}^H$ is quantized into $H$ sign bits and $H$ 7-bit magnitudes and these bits are packed into a $H$-bits word for signs and 7 $H$-bits word for magnitudes as illustrated in Algorithm 3. Then, the hash score is calculated via fast bitwise operations, such as bitwise XOR, AND, NOT, bit count, and bit shift instructions as shown in Algorithm 3. Finally, the top-$k$ candidates with highest hash scores are subsequently selected for sparse attention computation.

**Error-Aware Safeguard (EAS).**

A key advantage of BinaryPC's reconstruction-based design is its error signal:

$$||\mathbf{k} - \mathbf{h}\mathbf{P}||_2 \tag{8}$$

It naturally enables an error-aware safeguard mechanism integrated within the retrieval mechanism. Although hashing-based retrieval greatly reduces computation, approximating attention scores via binary codes incurs inevitable precision loss: for tokens with small reconstruction errors, the inner product approximation in Eq. (3) remains accurate, whereas large errors may degrade retrieval fidelity. As a

result, during the prefill phase, we compute the per-token reconstruction error via Eq. (8) and identify the tokens with the top-$m$ largest errors as a hard-to-hash set $\mathcal{S}_{\text{err}}$. During decoding, the final set of tokens selected for sparse attention computation is the union:

$$\mathcal{S}_{\text{attn}} = \mathcal{S}_{\text{hash}} \cup \mathcal{S}_{\text{err}}, \tag{9}$$

where $\mathcal{S}_{\text{hash}}$ denotes the top-$k$ candidates retrieved using the hash codes, ensuring that critical tokens are preserved and providing a robust safeguard against approximation errors.

Experimental results demonstrate that even with compact 64-bit codes , BinaryPC approaches or matches full attention accuracy across various long-context benchmarks, validating the effectiveness of our approach.

# 4. Experiments

We empirically validate that BinaryPC substantially reduces decoding cost for large language models while preserving task accuracy. In Section 4.1, we evaluate BinaryPC on short-, medium-, and long-context benchmarks, demonstrating that it consistently approaches or matches full-attention accuracy across diverse task categories and model families. In Section 4.2, we show that BinaryPC with a 2% attention budget yields significant throughput improvement. In Section 4.3, we conduct ablation study to analyze the effect of the EAS mechanism.

## 4.1. Accuracy Evaluation

**Setup.** We evaluate BinaryPC on four widely used large language models: Llama-3-8B, Llama-3.1-8B-Instruct (Grattafiori et al., 2024), Mistral-7B-Instruct-v0.3 (Jiang et al., 2023), and Qwen2.5-7B-Instruct-1M (Yang et al., 2025b). Experiments are organized into four parts: (1) Short-context: three tasks from LM-Eval-Harness (Gao et al., 2024) (GSM8K-CoT (Cobbe et al., 2021), MMLU-Flan-Cot-Fewshot (Hendrycks et al., 2021), and CoQA (Reddy et al., 2019)); (2) Medium-

*Table 2.* Short-context evaluation on three benchmarks from LM-Eval-Harness (Gao et al., 2024).

| Method | Token | GSM8K | COQA | MMLU | Avg. |
|---|---|---|---|---|---|
| **Llama-3-8B** | Full | 54.50 | 80.53 | 59.73 | 64.92 |
| Quest | 64 | 3.10 | 75.69 | 27.87 | 35.55 |
| MagicPIG | Default | 40.40 | 76.77 | 52.21 | 56.46 |
| Spotlight | 5% | 40.00 | 80.11 | 56.81 | 58.97 |
| BinaryPC | 5% | 42.60 | 79.63 | 55.70 | 59.31 |
| BinaryPC w/ OPC | 5% | 52.00 | 80.28 | 59.18 | **63.82** |
| **Llama-3.1-8B** | Full | 73.80 | 78.82 | 65.60 | 72.74 |
| Quest | 64 | 12.00 | 71.82 | 25.78 | 36.53 |
| MagicPIG | Default | 60.90 | 75.34 | 56.46 | 64.23 |
| BinaryPC | 5% | 69.20 | 78.74 | 64.50 | 70.81 |
| BinaryPC w/ OPC | 5% | 70.60 | 78.80 | 64.08 | **71.16** |

context: LongBench (Bai et al., 2024); (3) Long-context: InfiniteBench (Zhang et al., 2024) and LongBench v2 (Bai et al., 2025); (4) Scalability from 8K to 128K: RULER (Hsieh et al., 2024) and Needle-in-a-Haystack (NIAH) (Kamradt, 2023). Detailed dataset descriptions are provided in Appendix A.2.

**Baselines.** We compare BinaryPC and its offline calibrated variant, BinaryPC w/ OPC, against several representative sparse-attention approaches, including static selection methods (PyramidKV (Cai et al., 2025), CAKE (Qin et al., 2025), CompressKV (Lin et al., 2025)), query-aware selection (Quest (Tang et al., 2024)), and hashing-based methods (MagicPIG (Chen et al., 2025), Spotlight (Li et al., 2025)). For MagicPIG, the effective token budget is data-dependent and difficult to estimate. We therefore report results using its default configuration, whose effective budget is approximately 5–7% on three LM-Eval-Harness (Gao et al., 2024) tasks, while it remains around 2–3% on the other tasks, based on our empirical statistics. Because different methods adopt different token-budgeting strategies—some using a fixed percentage of the input length (e.g., 2%), while others use a fixed number of tokens (e.g., 1K) regardless of input length—we follow the most commonly used budget-setting strategy among the baselines when configuring the size of $\mathcal{S}_{\text{attn}}$ for BinaryPC to ensure a fair comparison. In addition, 10% of the token budget is reserved for EAS. Detailed baseline methods settings are provided in Appendix A.1. Among sparse attention methods, the best result is highlighted in **bold**, and the second-best is underlined.

**Short-context.** Table 2 compares BinaryPC and its variants against full-attention baselines and sparse attention methods on three short-context benchmarks from LM-Eval-Harness (Gao et al., 2024). Notably, Quest (Tang et al., 2024), which operates under a comparable token budget (64 tokens, exceeding 5% of the input length), exhibits a substantial performance drop. In contrast, BinaryPC maintains strong task performance. On Llama-3-8B, BinaryPC achieves an average score of 59.31, outperforming Mag-

*Table 3.* LongBench (Bai et al., 2024) evaluation results. Scores are averaged by task category.

| Methods | Token | S-Doc | M-Doc | Sum. | F-shot | Syn. | Code | Avg. |
|---|---|---|---|---|---|---|---|---|
| **Llama-3-8B** | Full | 18.25 | 9.62 | 18.05 | 68.99 | 4.99 | 67.66 | 30.63 |
| MagicPIG | Default | 16.96 | 9.18 | 16.67 | 68.20 | 5.17 | 66.55 | 29.78 |
| Spotlight | 2% | 18.26 | 9.19 | 18.13 | 69.27 | 5.06 | 65.19 | 30.31 |
| BinaryPC | 2% | 17.99 | 9.12 | 18.36 | 68.60 | 4.75 | 65.58 | 30.18 |
| BinaryPC w/ OPC | 2% | 17.61 | 9.70 | 17.89 | 69.02 | 4.65 | 66.87 | **30.35** |
| **Llama-3.1-8B** | Full | 43.40 | 46.46 | 28.95 | 69.25 | 55.50 | 59.57 | 49.64 |
| PyramidKV | 1K | 42.53 | 45.69 | 25.32 | 68.13 | 55.38 | 57.35 | 48.15 |
| Cake | 1K | 42.83 | 45.74 | 25.87 | 68.45 | 55.46 | 58.32 | 48.51 |
| CompressKV | 1K | 43.45 | 46.00 | 26.09 | 68.63 | 55.38 | 58.97 | 48.82 |
| Quest | 1K | 42.45 | 46.36 | 28.84 | 68.34 | 55.30 | 57.14 | 48.93 |
| MagicPIG | Default | 43.06 | 46.13 | 28.11 | 68.76 | 54.84 | 58.11 | 49.01 |
| BinaryPC | 2% | 43.59 | 46.33 | 29.00 | 69.16 | 55.25 | 58.61 | 49.50 |
| BinaryPC | 1K | 43.38 | 46.22 | 29.02 | 69.27 | 55.48 | 59.98 | 49.66 |
| BinaryPC w/ OPC | 2% | 43.58 | 46.24 | 28.74 | 68.93 | 55.38 | 59.42 | 49.50 |
| BinaryPC w/ OPC | 1K | 43.45 | 46.34 | 28.73 | 69.33 | 55.43 | 60.13 | **49.67** |
| **Mistral-7B** | Full | 38.63 | 39.66 | 28.63 | 70.74 | 52.00 | 60.24 | 47.31 |
| PyramidKV | 1K | 36.96 | 37.62 | 25.25 | 70.11 | 50.00 | 57.88 | 45.35 |
| Cake | 1K | 37.69 | 38.15 | 26.39 | 70.10 | 51.00 | 59.38 | 46.11 |
| CompressKV | 1K | 38.50 | 38.75 | 26.52 | 70.68 | 51.25 | 59.44 | 46.55 |
| Quest | 1K | 37.52 | 37.55 | 27.70 | 69.96 | 49.54 | 58.36 | 45.87 |
| MagicPIG | Default | 38.53 | 38.91 | 28.47 | 70.76 | 51.25 | 60.24 | 47.06 |
| BinaryPC | 2% | 38.32 | 38.66 | 28.70 | 70.49 | 52.00 | 60.32 | 47.07 |
| BinaryPC | 1K | 38.69 | 39.44 | 28.74 | 70.90 | 52.00 | 60.39 | 47.38 |
| BinaryPC w/ OPC | 2% | 39.23 | 38.88 | 28.60 | 71.04 | 51.50 | 59.78 | 47.24 |
| BinaryPC w/ OPC | 1K | 38.53 | 39.30 | 28.80 | 70.98 | 52.50 | 60.48 | **47.42** |

icPIG (Chen et al., 2025) and even the training-based Spotlight (Li et al., 2025). The offline variant further narrows the gap to full attention, reaching 63.82. On Llama-3.1-8B, BinaryPC achieves 70.81 average accuracy, substantially surpassing MagicPIG while closely tracking the full-attention baseline. These results demonstrate that BinaryPC effectively reduces attention computation without significantly degrading task-level quality in short-context scenarios.

**Medium-context.** We evaluate on LongBench (Bai et al., 2024), a bilingual benchmark comprising 16 datasets across six task categories. As shown in Table 3, on Llama-3-8B, BinaryPC outperforms MagicPIG and matches accuracy with Spotlight in a training-free manner, while Spotlight requires substantial per-model training to learn complex hashing mappings. On Llama-3.1-8B and Mistral-7B, BinaryPC with a 2% budget matches full-attention performance and outperforms all sparse-attention baselines. Notably, with a 1K budget, BinaryPC even surpasses full attention on both models. These findings demonstrate BinaryPC's strong generalization ability without model-specific tuning or auxiliary architectural modifications. Table 9 in the Appendix reports per-dataset performance across all 16 datasets for three model families, including additional results on Qwen2.5-7B. Appendix B.4 further shows that BinaryPC matches full attention on Llama-3-70B-Instruct while preserving the same relative hash-code memory overhead and similar single-layer attention latency.

**Long-context.** We evaluate BinaryPC on InfiniteBench and LongBench v2. InfiniteBench (Zhang et al., 2024) has an

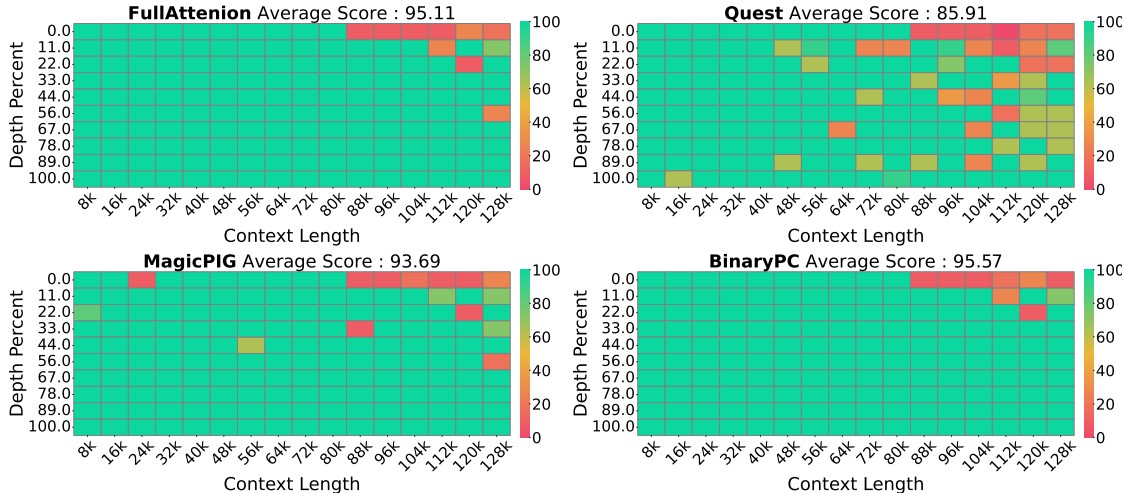

*Figure 4.* Evaluation on the NIAH benchmark (Kamradt, 2023) using Llama-3.1-8B-Instruct.

*Table 4.* RULER (Hsieh et al., 2024) scalability evaluation on Llama-3.1-8B-Instruct from 8K to 128K context length.

| Methods | Token | 8K | 16K | 32K | 64K | 128K | Avg. |
|---|---|---|---|---|---|---|---|
| **Llama-3.1-8B** | Full | 94.32 | 93.96 | 87.10 | 85.19 | 76.49 | 87.41 |
| TOPK | 2% | 91.53 | 91.37 | 88.80 | 83.39 | 73.62 | 85.74 |
| PyramidKV | 2K | 82.64 | 79.39 | 72.69 | 68.34 | 48.52 | 70.32 |
| Cake | 2K | 91.48 | 85.37 | 78.14 | 72.28 | 61.61 | 77.78 |
| CompressKV | 2K | 89.52 | 83.83 | 78.83 | 74.60 | 65.32 | 78.42 |
| Quest | 2K | 92.36 | 90.52 | 83.10 | 78.45 | 65.39 | 81.96 |
| MagicPIG | Default | 91.28 | 90.82 | 85.26 | 83.96 | 72.07 | 84.68 |
| BinaryPC | 2% | 91.67 | 90.86 | 87.73 | 84.36 | 73.23 | 85.57 |
| BinaryPC | 2K | 94.47 | 93.98 | 88.86 | 85.09 | 72.89 | **87.06** |
| BinaryPC w/ OPC | 2% | 90.56 | 91.15 | 87.36 | 84.14 | 73.27 | 85.29 |
| BinaryPC w/ OPC | 2K | 94.17 | 93.88 | 89.55 | 84.77 | 72.56 | 86.99 |

average sequence length exceeding 100K tokens, and we assess seven task types from this suite. LongBench v2 (Bai et al., 2025) stratifies results by both difficulty and context length. As shown in Table 1, BinaryPC delivers performance that is nearly equivalent to full attention across both benchmarks. Additional Qwen2.5-7B results in Appendix Table 8 further show consistent gains over sparse baselines, demonstrating that the long-context robustness generalizes across model families.

**Scalability from 8K to 128K.** We assess the scalability and stability of BinaryPC across context lengths from 8K to 128K tokens using the RULER (Hsieh et al., 2024) benchmark and NIAH (Kamradt, 2023). On RULER (shown in Table 4), BinaryPC consistently outperforms other sparse methods and maintains stable performance across scaling regimes. Detailed per-task RULER results across different context lengths are provided in Appendix B.9. As shown in Figure 1, BinaryPC achieves performance nearly identical to the oracle TOPK and closely aligns with full attention. In contrast, MagicPIG consistently underperforms the oracle TOPK baseline. For NIAH evaluation (Figure 4), BinaryPC matches full-attention retrieval accuracy across all needle

positions and context lengths, and even surpasses full attention in certain cases. More detailed NIAH results are included in the Appendix (Figures 10 and 9) for a comprehensive comparison of more baselines and BinaryPC variants across models and context lengths. Beyond retrieval accuracy, Appendix B.1 further shows that BinaryPC provides a better approximation to full attention outputs than MagicPIG under comparable token budgets, achieving consistently higher cosine similarity across context lengths from 8K to 64K. In summary, BinaryPC achieves superior performance compared to all baselines.

### 4.2. Efficiency Evaluation

**Setup.** All efficiency experiments were conducted on Llama-3.1-8B-Instruct (Grattafiori et al., 2024) using eight data-center GPUs, evaluating the offline-calibrated variant of BinaryPC. We employed the HuggingFace Transformers framework with pipeline parallelization and a pre-allocated static KV cache to maximize throughput. To assess model throughput at extended context lengths, we expanded positional encoding while disregarding output quality. Among hashing-based baselines, MagicPIG (Chen et al., 2025) uses hash codes exceeding 1000 bits and relies on GPU–CPU collaborative decoding, which substantially reduces throughput. As shown in Appendix Table 12, MagicPIG reaches only 0.42–0.50× the throughput of FlashAttention-2 across batch sizes and context lengths, whereas BinaryPC achieves 1.05–1.69× speedup under the same single-GPU setting. We therefore adopt FlashAttention-2 (Dao et al., 2022; Dao, 2024) as the primary efficiency baseline. We generated 64 consecutive tokens after a 32-step warm-up phase and averaged three runs per data point. To validate the generality of our approach across different hardware, we performed efficiency experiments on eight consumer-grade GPUs and analyzed the runtime of BinaryPC and its offline variant

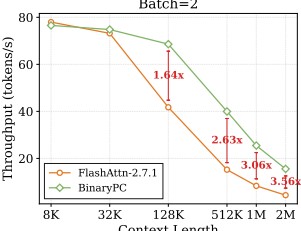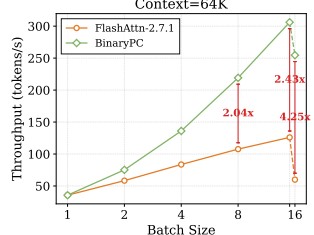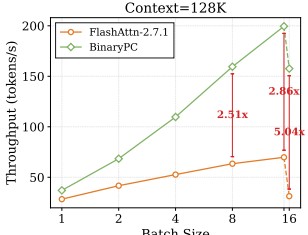

*Figure 5.* End-to-end throughput comparison. **Left two:** throughput across context lengths (8K–2M) for batch sizes of 1 and 2. **Right two:** throughput across batch sizes (1–16) at fixed context lengths of 64K and 128K tokens.

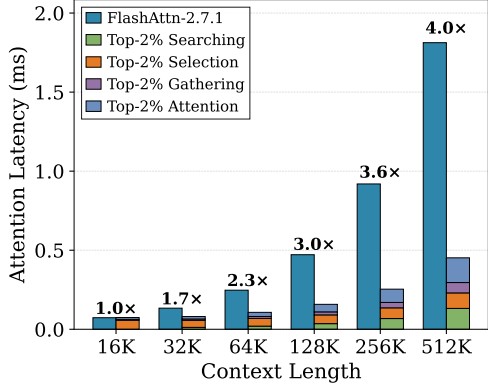

*Figure 6.* Attention layer execution time breakdown on Llama-3.1-8B-Instruct under the default BinVortex configuration.

during the prefill stage, as reported in Appendix B.2.

**Kernel Evaluation.** To fully exploit GPU hardware parallelism, we implement optimized CUDA kernels for binary hashing and hash score computation. Key vectors are transformed into 64-bit binary hash codes via constructed projection matrices. For hash score calculation, each 8-bit quantized query element is decomposed into 8 bit-planes, which are packed as 64-bit words. The hash score calculation in Eq. (3) is then computed by performing bitwise and bit count operations between each query bit-plane and the binary key hash via Algorithm 3. Figure 6 provides a detailed breakdown of the attention layer execution time across varying context lengths. The retrieval pipeline consists of three stages: (1) *Searching*, where a custom CUDA kernel computes the hash score between projected queries and binary key hash codes using bitwise and bit count operations; (2) *Selection*, which identifies the top-$k$ tokens via *torch.topk*; and (3) *Gathering*, which retrieves the corresponding key-value pairs using *torch.gather*. Appendix A.4 details how BinaryPC packs each 64-bit key hash into one `int64` and computes hash scores using XOR and population-count operations. Taking the 512K length setting as an example, these three stages introduce a combined overhead of 295 $\mu$s, but BinaryPC can reduce the FlashAttention kernel invocation time from 1.814 ms to 163 $\mu$s, corresponding to a 4.0× speedup for a single attention layer. The

overhead introduced by the retrieval stages is small relative to the total decoding latency, validating the efficiency of the proposed mechanism in long-sequence scenarios.

**End-to-end Throughput.** Figure 5 reports the end-to-end decoding throughput across context lengths from 8K to 2M tokens. Under small-batch settings, BinaryPC consistently outperforms the FlashAttention-2 baseline. For short contexts, BinaryPC attains comparable throughput to FlashAttention-2, while for long contexts it achieves up to 3.17× and 3.56× improvement, with the performance gap further widening as batch size and context length increase.

As the batch size increases at fixed context lengths of 64K and 128K, BinaryPC maintains an increasingly larger efficiency margin. Notably, on the data-center GPU setup, FlashAttention-2 exhibits a sharp throughput drop once the batch size exceeds 15, due to an internal heuristic that switches from the highly parallel Split-K kernel to the standard kernel. While BinaryPC also invokes FlashAttention-2 kernels internally, its smaller FlashAttention input size substantially mitigates this throughput degradation. At a batch size of 16, BinaryPC achieves 4.25× and 5.04× improvements at context lengths of 64K and 128K, respectively.

### 4.3. Ablation Study

**Effect of EAS.** We first conduct an EAS budget ablation on InfiniteBench (Zhang et al., 2024) using Llama-3.1-8B-Instruct with 100 samples per subtask, varying the EAS ratio over $\{0\%, 2\%, 5\%, 10\%, 15\%\}$. As shown in Table 5, EAS is essential for standard BinaryPC: without EAS, R.PK drops to 64.00 and the average score falls to 41.30, while a minimal 2% budget fully recovers R.PK to 99.00. This degradation arises because the passkey is highly dissimilar from the surrounding context, causing it to behave as an outlier during online projection construction and making it difficult to be captured with binary codes. Once EAS is activated, performance is insensitive to its budget ratio, with BinaryPC averages varying only from 47.42 to 47.63 across 2–15%. This indicates that EAS is a robust algorithmic component rather than a fragile hyperparameter. OPC provides complementary robustness: with offline projection

calibration, R.PK already reaches 99.00 even at 0% EAS, while EAS still yields modest gains on other tasks such as En.Dia (15.00 to 18.00). We therefore use 10% as the default EAS budget, as it achieves near-optimal performance, introduces negligible overhead due to static query-agnostic selection, and provides a comfortable margin above the 2% effectiveness threshold.

*Table 5.* Ablation study on the EAS budget ratio using InfiniteBench with Llama-3.1-8B-Instruct, evaluated with 100 samples per subtask.

| Methods | EAS | En.Sum | En.QA | En.MC | En.Dia | Zh.QA | Math.F | R.PK | Avg. |
|---|---|---|---|---|---|---|---|---|---|
| Full Attention | – | 32.85 | 27.41 | 72.00 | 20.00 | 36.17 | 48.00 | 99.00 | 47.92 |
| | 0% | 32.16 | 26.43 | 72.00 | 15.00 | 35.92 | 48.00 | 99.00 | 46.93 |
| | 2% | 31.85 | 26.32 | 72.00 | 18.00 | 36.18 | 48.00 | 99.00 | 47.34 |
| BinaryPC w/ OPC | 5% | 32.78 | 26.40 | 72.00 | 18.00 | 36.83 | 48.00 | 99.00 | 47.57 |
| | 10% | 32.44 | 26.43 | 72.00 | 18.00 | 36.67 | 48.00 | 99.00 | 47.51 |
| | 15% | 32.39 | 26.35 | 72.00 | 18.00 | 36.16 | 48.00 | 99.00 | 47.41 |
| | 0% | 30.97 | 24.33 | 72.00 | 17.00 | 32.83 | 48.00 | 64.00 | 41.30 |
| | 2% | 32.50 | 26.18 | 72.00 | 18.00 | 36.28 | 48.00 | 99.00 | 47.42 |
| BinaryPC | 5% | 32.48 | 25.53 | 72.00 | 20.00 | 36.38 | 48.00 | 99.00 | **47.63** |
| | 10% | 32.46 | 26.17 | 72.00 | 18.00 | 36.24 | 48.00 | 99.00 | 47.41 |
| | 15% | 32.15 | 26.03 | 72.00 | 19.00 | 36.39 | 48.00 | 99.00 | 47.51 |

**Hash bit length.** We further study the effect of hash bit length on NIAH (Kamradt, 2023) using Llama-3.1-8B-Instruct. As shown in Table 6, 32-bit hashing exhibits a notable drop at 104K context length, reducing the average score to 86.37. In contrast, 64-bit and 128-bit hashing perform similarly well, achieving average scores of 90.18 and 90.00, respectively. From an implementation perspective, 64-bit hash codes fit into a single `int64` word and naturally support efficient bitwise operations, offering the best trade-off between accuracy and efficiency.

*Table 6.* Ablation study on hash bit length using NIAH with Llama-3.1-8B-Instruct.

| Hash Bits | 88K | 96K | 104K | 112K | 120K | Avg. |
|---|---|---|---|---|---|---|
| 32-bit | 91.82 | 91.82 | 71.82 | 91.82 | 84.55 | 86.37 |
| 64-bit | 90.91 | 91.82 | 92.73 | 90.91 | 84.55 | **90.18** |
| 128-bit | 91.82 | 91.82 | 90.91 | 91.82 | 83.64 | 90.00 |

**OPC robustness under domain shift.** We next examine whether OPC depends on calibration-data diversity. To address this concern, we calibrate the OPC variant using only PG19 (Rae et al., 2020), a corpus of long-form literary text, and evaluate it on the full LongBench (Bai et al., 2024) benchmark spanning six diverse task categories, including code, synthetic reasoning, and multi-document QA, which are absent from the calibration data. As shown in Table 7, the PG19-only OPC variant achieves an average score of 30.43, comparable to both full attention (30.57) and the mixed-domain OPC variant (30.39). It also shows no systematic degradation on out-of-domain tasks such as Code (67.06 vs. 67.77) or Synthetic (4.71 vs. 4.78), suggesting that calibration-data diversity has only marginal impact on OPC performance.

*Table 7.* Ablation study on OPC robustness under calibration-domain shift using LongBench (Bai et al., 2024).

| Methods | S-Doc | M-Doc | Sum. | F-shot | Syn. | Code | Avg. |
|---|---|---|---|---|---|---|---|
| Full Attention | 17.97 | 9.57 | 17.97 | 69.15 | 4.78 | 67.77 | 30.57 |
| BinaryPC w/ OPC (mix) | 17.78 | 9.66 | 17.50 | 69.10 | 5.44 | 66.66 | 30.39 |
| BinaryPC w/ OPC (PG19 only) | 18.26 | 9.47 | 17.78 | 68.92 | 4.71 | 67.06 | **30.43** |

## 5. Conclusion

We presented BinaryPC, a training-free and data-aware hashing-based sparse attention for efficient long-context LLM inference. By constructing compact hash codes that explicitly preserve the structural information of data under the hashing projection, BinaryPC enables efficient and accurate token retrieval using GPU-friendly bit-parallel operations, substantially reducing the computational cost of attention during decoding. An error-aware safeguard further ensures robust recall by explicitly retaining hard-to-hash tokens. Extensive experiments across multiple model families and benchmarks demonstrate that BinaryPC preserves accuracy while reducing effective KV retrieval and attention computation. Compared to sparse attention baselines, BinaryPC consistently achieves superior task performance, while exhibiting strong scalability and stability from 8K to 128K context lengths. Moreover, BinaryPC delivers significant decoding speedups on modern GPUs without requiring model-specific training or architectural modifications. Overall, these results highlight BinaryPC as a lightweight, practical, and scalable hashing-based approach for long-context LLM inference.

## Acknowledgements

This work was supported by the National Key Research and Development Program of China (No. 2025YFE0113500), the National Science Fund for Distinguished Young Scholars (No. 62525605), and the National Natural Science Foundation of China (No. U25B2066, No. 62506313).

## Impact Statement

The deployment of long-context large language models (LLMs) is frequently constrained by substantial computational demands during decoding, limiting their accessibility for real-world applications that require processing extended sequences. This work introduces BinaryPC, a training-free sparse attention that addresses these constraints by enabling efficient KV cache retrieval through compact 64-bit binary hashing. By leveraging data-aware hashing projections and GPU-optimized bitwise operations, BinaryPC significantly reduces attention computation while preserving task accuracy.

The societal implications of BinaryPC are multifaceted. By enabling efficient long-context inference on existing hardware, this work democratizes access to advanced LLM capabilities, empowering smaller organizations, independent researchers, and practitioners in resource-constrained settings. The ability to achieve up to $3.17\times$ throughput improvements and $5.04\times$ speedup in Flash Attention fallback scenarios translates directly to reduced energy consumption and lower operational costs for large-scale deployments. This contributes to the broader sustainability goals of AI by minimizing the environmental footprint associated with serving long-context workloads. Furthermore, the training-free nature of BinaryPC eliminates the need for model-specific optimization or extensive calibration data, making it readily applicable across diverse model families without additional engineering overhead. This model agnosticism promotes equitable access to efficient inference techniques, ensuring that advancements in sparse attention are not confined to well-resourced institutions with the capacity for per-model fine-tuning.

No specific ethical concerns or societal risks are associated with the proposed method. BinaryPC does not alter the fundamental capabilities or outputs of the underlying LLMs; it solely optimizes the computational efficiency of attention mechanisms. As such, it provides a pathway toward more equitable and responsible AI deployment, ensuring that advances in long-context LLM inference are accessible and sustainable across diverse sectors and applications.

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

# A. Implementation Details

## A.1. Baseline Methods

**TOPK (Oracle).** TOPK selects the top-$k$ keys using exact full-precision attention scores under a fixed 2% token budget, then restricts attention to this selected subset.

**PyramidKV.** PyramidKV employs a pyramid-shaped budget allocation strategy where shallow layers retain more KV pairs than deeper layers. We set the local window size to 8, pooling kernel size to 5 with average pooling, and the steepness parameter $\beta = 20$. The first two layers remain dense to preserve critical early-context information.

**Cake.** Cake dynamically allocates layer-wise budgets based on attention statistics from the prefill phase. We set the local window size to 8. Budget allocation parameters $\tau_1$ and $\tau_2$ follow the settings for Llama-3.1-8B-Instruct and Mistral-7B-Instruct-v0.3, with the eviction parameter $\gamma = 200.0$. The first two layers remain dense. Evaluations are conducted only on the models where detailed parameters are publicly available.

**CompressKV.** CompressKV utilizes a subset of important attention heads to estimate token importance. We set the local window size to 8 and pooling kernel size to 5. The number of active heads $k$ is set to 4. Head and layer weights are pre-computed offline using a calibration set. The first two layers remain dense. Evaluations are conducted only on the models where detailed parameters are publicly available.

**Quest.** Quest performs query-aware token selection via chunk-based importance estimation. We set the chunk size to 16. During decoding, top-$k$ chunks are identified based on the maximum attention weight within each chunk. The first two layers remain dense.

**MagicPIG.** We adopt the default configuration, where LSH retrieval uses $K = 10$ hash bits and $L = 150$ tables, with a local window size of 64 and 4 sink tokens. The collision threshold is set to 2 and error correction is enabled via the anns_es mode. The first and middle layers remain dense.

**Spotlight.** We adopt the official implementation with a 2% token budget and a minimum retention of 20 tokens. The first two layers remain dense. Due to the training requirement, we conduct evaluations only on Llama-3-8B using the publicly released weights.

## A.2. Dataset Details

We evaluate BinaryPC across four context regimes using diverse benchmarks that comprehensively assess task accuracy, retrieval capability, and scalability.

**Short-context: LM-Eval-Harness.** We employ three representative tasks from LM-Eval-Harness (Gao et al., 2024): GSM8K-CoT (Cobbe et al., 2021) for mathematical reasoning with chain-of-thought prompting, MMLU-Flan-CoT-Fewshot (Hendrycks et al., 2021) for multi-task language understanding, and CoQA (Reddy et al., 2019) for conversational question answering. Following the standard few-shot protocol, we sample 1000 instances per subtask and report accuracy for GSM8K and MMLU, and F1 for CoQA.

**Medium-context: LongBench.** LongBench (Bai et al., 2024) is the first benchmark for bilingual, multitask, and comprehensive assessment of long context understanding capabilities. Featuring both Chinese and English languages, LongBench contains 16 datasets spanning six task categories: Single-Document QA (NrtvQA, Qasper, MF-en), Multi-Document QA (HotpotQA, 2WikiMQA, Musique), Summarization (GovReport, QMSum, MultiNews), Few-shot Learning (TREC, TriviaQA, SAMSum), Synthetic Retrieval (PCount, PRe), and Code Completion (Lcc, RB-P). The benchmark includes 14 English tasks, 5 Chinese tasks, and 2 code tasks, with the average length of most tasks ranging from 5K to 15K tokens.

**Long-context: InfiniteBench and LongBench v2.** InfiniteBench (Zhang et al., 2024) is a cutting-edge benchmark tailored for evaluating the capabilities of language models to process, understand, and reason over super long contexts. It is designed to push the boundaries of language models by testing them against a context length of 100K+, which is 10 times longer than traditional datasets. We select seven task categories for evaluation: English summarization (En.Sum), English question answering (En.QA), English multiple-choice (En.MC), speaker identification (En.Dia), Chinese question answering (Zh.QA), special-integer retrieval (Math.F), and passkey retrieval (R.PK). LongBench v2 (Bai et al., 2025) is designed to assess the ability of LLMs to handle long-context problems requiring deep understanding and reasoning across real-world multitasks. It consists of 503 challenging multiple-choice questions with context lengths ranging from 8K to 2M words, across six major task categories: single-document QA, multi-document QA, long in-context learning, long-dialogue history understanding,

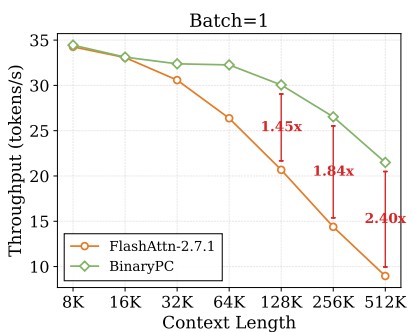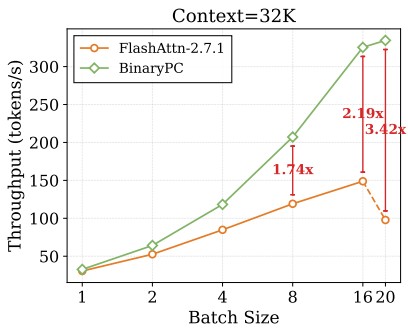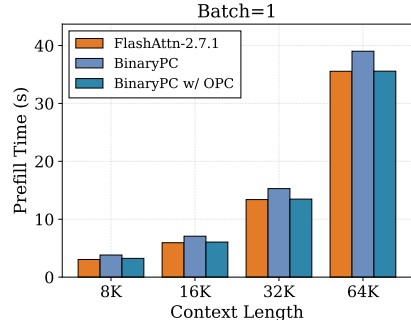

*Figure 7.* Efficiency evaluation on consumer-grade GPUs. Left: Decoding throughput across context lengths from 8K to 512K tokens at batch size 1. Middle: Throughput scaling with batch sizes from 1 to 20 at a fixed 32K context length. Right: Prefill stage latency comparison between BinVortex variants and FlashAttention-2.

*Table 8.* Performance on InfiniteBench (Zhang et al., 2024) (left) and LongBench v2 (Bai et al., 2025) (right).

| Methods | Token | InfiniteBench | | | | | | | | LongBench v2 | | | | | |
| | | En.Sum | En.QA | En.MC | En.Dia | Zh.QA | Math.F | R.PK | Avg. | Easy | Hard | Short | Medium | Long | Avg. |
|---|---|---|---|---|---|---|---|---|---|---|---|---|---|---|---|
| **Qwen2.5-7B** | Full | 34.02 | 12.14 | 68.12 | 19.00 | 9.90 | 40.29 | 34.58 | 31.15 | 34.9 | 27.7 | 37.8 | 26.5 | 25.9 | 30.4 |
| PyramidKV | 2K | 27.48 | 11.17 | 67.69 | 10.00 | 9.20 | 39.43 | 34.58 | 28.50 | 35.4 | 27.3 | 37.2 | 26.0 | 27.8 | 30.4 |
| Quest | 2K | 15.57 | 11.60 | 67.69 | 13.00 | 10.56 | 40.00 | 34.58 | 27.57 | 31.8 | 25.7 | 37.8 | 22.3 | 23.1 | 28.0 |
| MagicPIG | Default | 27.31 | 11.35 | 69.00 | 14.00 | 10.45 | 29.71 | 10.17 | 24.57 | 32.8 | 25.7 | 31.7 | 28.4 | 23.1 | 28.4 |
| BinaryPC | 2% | 31.40 | 11.43 | 67.25 | 14.00 | 10.45 | 39.71 | 34.58 | **29.83** | 35.9 | 28.3 | 38.9 | 27.4 | 25.9 | **31.2** |
| BinaryPC w/ OPC | 2% | 31.07 | 11.57 | 69.00 | 15.50 | 10.22 | 33.43 | 34.41 | 29.31 | 35.4 | 27.3 | 38.9 | 27.0 | 23.1 | 30.4 |

code repository understanding, and long structured data understanding. Results are stratified by difficulty (Easy/Hard) and context length (Short/Medium/Long).

**Scalability from 8K to 128K: RULER and Needle-in-a-Haystack.** RULER (Hsieh et al., 2024) generates synthetic examples to evaluate long-context language models with configurable sequence length and task complexity. It consists of 13 complex tasks across 4 task categories, including retrieval, multi-hop tracing, aggregation, and question answering, evaluating long-context capabilities beyond simple in-context recall. We configure test sets scaling from 8K to 128K tokens to systematically evaluate performance degradation as context length increases. Needle-in-a-Haystack (NIAH) (Kamradt, 2023) is a long-context retrieval benchmark that evaluates LLM performance with extended context windows, where relevant information is distributed at varying depths.

### A.3. Procedure Overview of BinaryPC

Figure 8 provides an intuitive overview of the BinaryPC procedure. Unlike LSH, which partitions the space using data-independent random hyperplanes, BinaryPC derives hashing directions from the principal directions of the key vectors, aligning partitions with the intrinsic data structure and enabling more faithful similarity preservation.

BinaryPC iteratively solves the reconstruction objective $\min_{\mathbf{H},\mathbf{P}} \|\mathbf{K} - \mathbf{H}\mathbf{P}\|_F^2$ one bit at a time. Starting from the residual $\mathbf{R} = \mathbf{K}$, each iteration maps every data point $\mathbf{k}_i \in \mathbf{K}$ to a binary sign $u_i \in \{+1, -1\}$ through space partitioning and computes a shared projection vector $\mathbf{v}$. For a fixed binary vector $\mathbf{u}$, the optimal projection vector has the closed-form solution $\mathbf{v} = \mathbf{u}^\top \mathbf{R}/N$, where $N$ is the number of data points. This creates a rank-1 approximation $\mathbf{u}\mathbf{v}^\top$. The algorithm then updates the residual as $\mathbf{R} \leftarrow \mathbf{R} - \mathbf{u}\mathbf{v}^\top$, which is passed to the next iteration as the new signal.

### A.4. CUDA Optimizations on Bitwise Operations

We implement three CUDA kernels for BinaryPC.

**Hashing kernel.** Each key vector ($D = 128$, BF16) is projected to 64 binary values and directly packed into a single `int64` word. Let $\mathbf{p} = \mathbf{q}\mathbf{P}^\top$ for notation simplicity.

**Packing kernel.** Each 64-entry quantized vector $\mathbf{p}$ with values in $[-127, 127]$ is packed into one `int64` $\mathbf{s}$ storing the signs and seven `int64` values $\mathbf{m}[j]$ storing the magnitude bits.

*Table 9.* LongBench (Bai et al., 2024) evaluation across 16 datasets.

| Method | Token | Single-Document QA | | | Multi-Document QA | | | Summarization | | | Few-shot Learning | | | Synthetic | | Code | | Avg |
|---|---|---|---|---|---|---|---|---|---|---|---|---|---|---|---|---|---|---|
| | | NrtvQA | Qasper | MF-en | HotpotQA | 2WikiMQA | Musique | GovReport | QMSum | MultiNews | TREC | TriviaQA | SAMSum | PCount | PRe | Lcc | RB-P | |
| **Llama-3-8B** | Full | 16.94 | 14.18 | 23.62 | 9.97 | 11.75 | 7.15 | 29.85 | 23.05 | 1.26 | 70.50 | 91.20 | 45.26 | 1.70 | 8.29 | 70.83 | 64.49 | 30.63 |
| MagicPIG | Default | 15.54 | 14.20 | 21.13 | 9.45 | 11.30 | 6.78 | 25.84 | 23.45 | 0.72 | 69.50 | 90.97 | 44.13 | 1.95 | 8.38 | 70.06 | 63.04 | 29.78 |
| Spotlight | 2% | 15.44 | 14.94 | 24.39 | 9.12 | 11.81 | 6.64 | 29.34 | 22.47 | 2.57 | 72.00 | 91.20 | 44.61 | 2.50 | 7.62 | 67.37 | 63.01 | 30.31 |
| BinaryPC | 2% | 15.39 | 14.10 | 21.25 | 9.27 | 12.27 | 6.99 | 29.92 | 23.27 | 1.54 | 70.50 | 91.20 | 44.69 | 2.00 | 7.04 | 69.85 | 63.61 | 30.18 |
| BinaryPC w/ OPC | 2% | 16.92 | 13.09 | 22.81 | 9.60 | 12.01 | 7.48 | 29.90 | 22.81 | 0.97 | 70.50 | 91.20 | 45.36 | 1.62 | 7.67 | 70.28 | 63.45 | **30.35** |
| **Llama-3.1-8B** | Full | 29.56 | 44.70 | 55.93 | 57.82 | 48.95 | 32.61 | 34.45 | 25.51 | 26.88 | 72.50 | 91.15 | 44.10 | 11.50 | 99.50 | 62.97 | 56.16 | 49.64 |
| PyramidKV | 1K | 30.45 | 41.75 | 55.38 | 57.13 | 49.25 | 30.68 | 26.69 | 23.85 | 25.41 | 70.50 | 91.18 | 42.71 | 11.25 | 99.50 | 61.84 | 52.85 | 48.15 |
| Cake | 1K | 29.99 | 41.69 | 56.82 | 57.16 | 48.06 | 32.00 | 27.66 | 24.31 | 25.63 | 70.50 | 91.52 | 43.33 | 10.92 | 100.00 | 61.53 | 55.10 | 48.51 |
| CompressKV | 1K | 29.61 | 43.52 | 57.21 | 57.60 | 48.47 | 31.93 | 28.18 | 24.43 | 25.67 | 71.00 | 91.18 | 43.70 | 11.25 | 99.50 | 62.59 | 55.34 | 48.82 |
| Quest | 1K | 30.08 | 43.60 | 53.66 | 57.45 | 48.75 | 32.88 | 34.46 | 25.22 | 26.83 | 72.25 | 89.99 | 42.78 | 11.10 | 99.50 | 60.85 | 53.44 | 48.93 |
| MagicPIG | Default | 30.47 | 42.57 | 56.13 | 57.26 | 48.33 | 32.80 | 32.94 | 25.16 | 26.24 | 72.50 | 90.86 | 42.93 | 11.17 | 98.50 | 60.80 | 55.43 | 49.01 |
| BinaryPC | 2% | 29.09 | 45.10 | 56.57 | 58.14 | 48.84 | 32.00 | 34.92 | 25.13 | 26.96 | 72.50 | 90.92 | 44.06 | 11.00 | 99.50 | 62.09 | 55.12 | **49.50** |
| BinaryPC w/ OPC | 2% | 30.06 | 43.81 | 56.87 | 58.31 | 48.46 | 31.95 | 34.57 | 25.20 | 26.45 | 72.50 | 91.12 | 43.16 | 11.25 | 99.50 | 62.71 | 56.12 | **49.50** |
| **Mistral-7B** | Full | 27.35 | 38.03 | 50.50 | 51.41 | 38.89 | 28.69 | 34.05 | 25.32 | 26.51 | 76.00 | 88.89 | 47.33 | 6.50 | 97.50 | 59.03 | 60.97 | 47.31 |
| PyramidKV | 1K | 26.18 | 34.89 | 49.80 | 49.78 | 36.98 | 26.09 | 27.04 | 23.54 | 25.18 | 74.50 | 89.86 | 45.97 | 4.00 | 96.00 | 57.63 | 58.12 | 45.35 |
| Cake | 1K | 26.74 | 36.04 | 50.28 | 50.29 | 37.07 | 27.08 | 28.56 | 24.43 | 26.17 | 74.00 | 89.31 | 47.00 | 5.50 | 96.50 | 58.74 | 60.01 | 46.11 |
| CompressKV | 1K | 27.26 | 37.10 | 51.14 | 50.10 | 38.20 | 27.95 | 28.98 | 24.80 | 25.79 | 76.00 | 89.57 | 46.46 | 5.50 | 97.00 | 58.66 | 60.22 | 46.55 |
| Quest | 1K | 26.35 | 36.80 | 49.42 | 47.56 | 38.15 | 26.95 | 32.26 | 24.44 | 26.40 | 75.50 | 89.08 | 45.29 | 5.32 | 93.75 | 58.00 | 58.71 | 45.87 |
| MagicPIG | Default | 26.78 | 37.51 | 51.31 | 50.76 | 37.80 | 28.18 | 33.40 | 25.67 | 26.33 | 76.00 | 89.09 | 47.18 | 6.00 | 96.50 | 59.64 | 60.84 | 47.06 |
| BinaryPC | 2% | 26.84 | 37.65 | 50.46 | 50.65 | 37.90 | 27.42 | 34.58 | 24.89 | 26.64 | 75.50 | 89.41 | 46.56 | 6.50 | 97.50 | 58.84 | 61.80 | **47.07** |
| BinaryPC w/ OPC | 2% | 27.53 | 39.57 | 50.60 | 50.30 | 37.64 | 28.69 | 34.27 | 25.12 | 26.40 | 76.00 | 89.86 | 47.27 | 5.50 | 97.50 | 59.10 | 60.45 | **47.24** |
| **Qwen2.5-7B** | Full | 28.80 | 47.88 | 50.38 | 58.74 | 54.09 | 33.80 | 33.37 | 23.51 | 24.28 | 76.50 | 84.07 | 44.61 | 7.00 | 100.00 | 46.19 | 39.24 | 47.03 |
| PyramidKV | 1K | 28.29 | 44.58 | 49.01 | 56.87 | 52.69 | 31.66 | 25.96 | 20.71 | 21.99 | 74.50 | 84.08 | 44.11 | 7.00 | 100.00 | 43.94 | 35.32 | 45.04 |
| Quest | 1K | 29.54 | 46.86 | 48.97 | 58.07 | 53.96 | 33.69 | 33.17 | 22.82 | 23.81 | 75.50 | 82.13 | 43.48 | 7.50 | 89.50 | 45.22 | 36.24 | 45.65 |
| MagicPIG | Default | 28.96 | 42.84 | 45.61 | 52.92 | 51.06 | 29.27 | 30.95 | 22.63 | 22.65 | 73.50 | 82.97 | 44.89 | 6.50 | 96.50 | 44.39 | 38.05 | 44.61 |
| BinaryPC | 2% | 29.94 | 45.99 | 49.60 | 58.01 | 53.93 | 32.23 | 33.50 | 23.35 | 24.72 | 77.00 | 85.50 | 44.28 | 7.50 | 100.00 | 42.01 | 36.14 | **46.48** |
| BinaryPC w/ OPC | 2% | 28.35 | 46.52 | 49.79 | 57.10 | 53.39 | 32.12 | 34.20 | 23.36 | 24.60 | 77.00 | 85.90 | 44.25 | 8.00 | 100.00 | 39.94 | 36.20 | 46.30 |

*Table 10.* Average cosine similarity between sparse and full attention outputs at the first decoding step on Llama-3.1-8B-Instruct.

| Method | 8K | 16K | 32K | 64K |
|---|---|---|---|---|
| BinaryPC (2.0%) | 0.9835 | 0.9811 | 0.9870 | 0.9895 |
| MagicPIG (2.2%) | 0.9388 | 0.9651 | 0.9595 | 0.9725 |

**Scoring kernel.** Using one `int64` hash code and the eight packed `int64` values ($\mathbf{s}$ and $\mathbf{m}[j]$), this kernel computes hash scores for all query-key pairs per head. Under this 8-`int64` representation, each $p_i$ is expressed as

$$p_i = s_i \sum_{j=1}^{7} 2^j m[j]_i. \tag{10}$$

The inner product between $\mathbf{p}$ and $\mathbf{h}$ in Eq. (3) can be written as

$$\sum_{i=1}^{64} h_i s_i \sum_{j=1}^{7} 2^j m[j]_i, \tag{11}$$

which can be arranged as

$$\sum_{j=1}^{7} 2^j \sum_{i=1}^{64} h_i s_i m[j]_i. \tag{12}$$

Since each $h_i$ is either $+1$ or $-1$, we use $\mathbf{s}$ XOR $\mathbf{h}$ to determine all $h_i s_i$ in a single CUDA instruction; denote the resulting `int64` data as $\mathbf{w}$. Then, each $\sum_{i=1}^{64} h_i s_i m[j]_i$ can be computed via

$$\text{\_\_popcll}(\mathbf{m}[j] \,\&\, \mathbf{w}) - \text{\_\_popcll}(\mathbf{m}[j] \,\&\, (\sim\mathbf{w})). \tag{13}$$

This counts the number of positive $(h_i s_i m[j]_i)$ contributions and subtracts the number of negative ones. To optimize memory I/O, the eight `int64` values are shared within each CUDA block to reduce data movement.

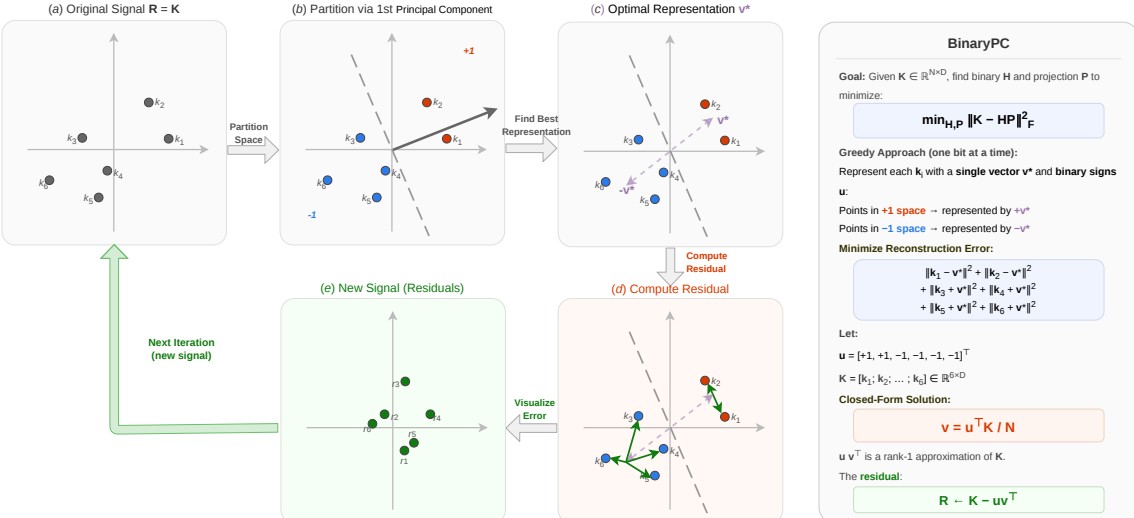

*Figure 8.* Procedure overview of BinaryPC. Each iteration partitions the current residual into binary signs, computes one shared projection component, removes the corresponding rank-1 approximation, and passes the remaining residual to the next iteration to generate the next bit.

## B. Additional Experiment Results

### B.1. Binary Representation and Approximation Quality

BinaryPC does not simply binarize vectors by taking the sign. Instead, it greedily minimizes the reconstruction objective $\|\mathbf{K} - \mathbf{HP}\|_F^2$ via iterative rank-1 binary decomposition. The rows of the projection matrix $\mathbf{P}$ represent components, and each key vector is represented as a weighted sum of these components, where the weights are $-1$ or $+1$. Therefore, the product $\mathbf{HP}$ reconstructs both directional and magnitude information.

**Approximation error of K.** As shown in Figure 3, as the number of binary principal components increases, the residual $\|\mathbf{K} - \mathbf{HP}\|_F$ decreases progressively, demonstrating the effectiveness of our reconstruction-based approach.

**Retrieval quality via Needle-in-a-Haystack.** Figures 4, 9, and 10 provide retrieval-accuracy comparisons across needle depths and context lengths. BinaryPC achieves near-perfect retrieval across all needle depths and context lengths, whereas competing methods exhibit systematic failures at specific depth ranges.

**Approximation error of attention output.** We measure the average cosine similarity between sparse and full attention outputs at the first decoding step on Llama-3.1-8B-Instruct. As shown in Table 10, under comparable token budgets of approximately 2%, BinaryPC consistently achieves higher cosine similarity than MagicPIG across all context lengths. This demonstrates that BinaryPC's reconstruction-based hashing provides superior quality for token selection.

### B.2. Additional Efficiency Evaluation

To validate the generality of BinaryPC across different hardware configurations, we conduct additional efficiency experiments on eight consumer-grade GPUs and analyze the runtime during the prefill stage.

**Decoding Throughput.** Figure 7 (left and middle) reports the decoding throughput on consumer-grade GPUs. Under batch size 1, BinaryPC achieves comparable throughput to FlashAttention-2 at shorter context lengths, while attaining up to $2.4\times$ speedup at 512K tokens. At a fixed 32K context length, BinaryPC achieves $2.19\times$ improvement at batch size 16. On this hardware, when batch size exceeds 16, FlashAttention-2 switches from the highly parallel Split-K kernel to the standard kernel, resulting in throughput degradation. At batch size 20, BinaryPC achieves $3.42\times$ speedup. These results confirm the generality of BinaryPC across different hardware configurations.

**Prefill Stage Analysis.** Figure 7 (right) presents the runtime breakdown during the prefill stage. The prefill phase computes attention over the entire input context and constructs the binary hash codes for subsequent decoding. The offline calibrated variant (BinaryPC w/ OPC) achieves nearly identical prefill latency to FlashAttention-2, as it leverages pre-calibrated

*Table 11.* Ultra-large-scale evaluation of BinaryPC. The upper block reports LongBench (Bai et al., 2024) category-level accuracy on Llama-3-70B-Instruct (Grattafiori et al., 2024); the lower block reports single-attention-layer latency.

| Accuracy on LongBench | | | | | | |
|---|---|---|---|---|---|---|
| Method | S-Doc | M-Doc | Sum. | F-shot | Syn. | Code |
| Full Attention-70B | 42.18 | 47.51 | 27.31 | 70.83 | 38.50 | 54.24 |
| BinaryPC-70B | 42.06 | 47.78 | 26.98 | 70.96 | 38.25 | 55.22 |

| Single-attention-layer latency (ms) | | | | | | |
|---|---|---|---|---|---|---|
| Method | Budget | 32K | 64K | 128K | 256K | Avg. |
| Full Attention-8B | Full | 0.181 | 0.343 | 0.667 | 1.287 | 0.620 |
| Full Attention-70B | Full | 0.181 | 0.343 | 0.654 | 1.289 | 0.617 |
| BinaryPC-8B | 2048 | 0.245 | 0.243 | 0.246 | 0.242 | 0.244 |
| BinaryPC-70B | 2048 | 0.243 | 0.243 | 0.243 | 0.245 | 0.244 |

projection matrices without additional computation during inference. In contrast, the online variant (BinaryPC) incurs modest overhead for computing projection matrices from the input context. This one-time cost is amortized over multiple decoding steps, making it negligible for generation tasks with substantial output lengths.

### B.3. Additional Long-context Evaluation

Table 8 reports additional results on InfiniteBench (Zhang et al., 2024) and LongBench v2 (Bai et al., 2025) using Qwen2.5-7B-Instruct-1M (Yang et al., 2025b). BinaryPC consistently outperforms all sparse attention baselines on both benchmarks, achieving near-full-attention accuracy without model-specific tuning.

### B.4. Ultra-Large-Scale Model Evaluation

**Accuracy.** We further evaluate BinaryPC on Llama-3-70B-Instruct (Grattafiori et al., 2024) using LongBench (Bai et al., 2024). As shown in Table 11, BinaryPC matches the accuracy of full attention across LongBench task categories, demonstrating full-attention-level accuracy at the 70B scale.

**Memory overhead.** The additional memory introduced by BinaryPC comes from binary hash codes and projection matrices. Each hash code uses 8 bytes per token per KV head when stored as an `int64`, while the $128 \times 64$ projection matrix is negligible for long sequences. In comparison, the BF16 KV cache uses 512 bytes per token per KV head, so the hash-code overhead is only $1/64$, or approximately 1.56%, of the KV-cache size. Moving from Llama-3.1-8B-Instruct to Llama-3-70B-Instruct, the total number of KV-head instances across layers increases by $2.5\times$ ($8 \times 32$ to $8 \times 80$), while model size grows by $8.75\times$. Thus, the absolute BinaryPC memory overhead grows more slowly than model size, and the relative overhead with respect to the KV cache remains unchanged.

**Efficiency.** From Llama-3.1-8B-Instruct to Llama-3-70B-Instruct, the GQA group size doubles from 4 to 8, while the number of KV heads per layer remains unchanged. Since both regular attention and BinaryPC are primarily bottlenecked by KV-cache access and lightweight hashing overhead, the single-attention-layer latency remains similar across the two model scales. Table 11 reports this comparison using a 2048-token BinaryPC budget. Although the model size increases by $8.75\times$, the number of layers, and therefore the total attention runtime, increases by only $2.5\times$ from 32 to 80 layers.

### B.5. Efficiency Comparison with MagicPIG

Due to kernel limitations in MagicPIG (Chen et al., 2025), which relies on GPU-CPU collaborative decoding and is incompatible with multi-GPU parallelization, we conduct a direct comparison on a single data-center GPU using Llama-3.1-8B-Instruct (Grattafiori et al., 2024). As shown in Table 12, MagicPIG consistently achieves lower throughput than the FlashAttention-2 baseline across all tested configurations, with speedup ratios ranging from $0.42\times$ to $0.50\times$. In contrast, BinaryPC surpasses FlashAttention-2 in all settings, achieving speedups of $1.05\times$ to $1.69\times$. The performance gap widens as batch size increases: at 64K context length with batch size 4, BinaryPC achieves 148.46 tokens/s compared to MagicPIG's 42.05 tokens/s, representing a $3.53\times$ improvement over MagicPIG.

*Table 12.* Throughput comparison (tokens/s) between MagicPIG (Chen et al., 2025) and BinaryPC on a single data-center GPU. The gray text in brackets denotes batch size. Gain is computed relative to FlashAttention-2 (Dao, 2024).

| Context | FlashAttn-2.7.1 | MagicPIG | Gain | BinaryPC | Gain |
|---|---|---|---|---|---|
| 32K (1) | 45.31 | 20.28 | 0.45× | **47.61** | 1.05× |
| 32K (2) | 79.77 | 39.59 | 0.50× | **91.66** | 1.15× |
| 32K (4) | 123.98 | 57.06 | 0.46× | **167.72** | 1.35× |
| 64K (1) | 38.57 | 16.38 | 0.42× | **45.95** | 1.19× |
| 64K (2) | 62.40 | 29.09 | 0.47× | **84.64** | 1.36× |
| 64K (4) | 87.62 | 42.05 | 0.48× | **148.46** | 1.69× |

*Table 13.* Throughput comparison (tokens/s) of additional sparse-attention baselines on Llama-3.1-8B-Instruct using eight consumer-grade GPUs. FlashAttention-2 is measured with StaticCache, while other baselines follow their official implementations.

| Method | Token Budget | 16K | 32K | 64K | 128K |
|---|---|---|---|---|---|
| FlashAttention-2 | Full KV | 30.97 | 29.94 | 26.57 | 21.31 |
| CakeKV | 2,048 | 28.01 | 27.92 | 24.56 | 20.47 |
| PyramidKV | 2,048 | 28.88 | 28.30 | 27.74 | 28.14 |
| CompressKV | 2,048 | 28.61 | 28.66 | 28.30 | 29.01 |
| BinaryPC | 2,048 | 30.13 | 28.60 | 28.12 | 27.66 |

## B.6. Efficiency of Additional Baselines

The runtime efficiency results for the most relevant hashing-based baseline, MagicPIG (Chen et al., 2025), are reported in Table 12. As shown there, MagicPIG consistently achieves lower throughput than both FlashAttention-2 (Dao, 2024) and BinaryPC across all tested sequence lengths and batch sizes. In contrast, Spotlight (Li et al., 2025) does not provide publicly reproducible efficiency benchmarking code. Although CUDA kernels are released, no runnable instructions or examples are provided, making a fair latency comparison difficult.

Accordingly, we use FlashAttention-2 as the efficiency baseline for evaluating practical decoding efficiency, since it provides a strong and reproducible reference. Nevertheless, we made our best effort to further profile additional sparse-attention baselines for a more comprehensive comparison. Some methods could not be benchmarked without substantial engineering effort. The results we were able to obtain are shown in Table 13. We report throughput (tokens/s), which is inversely proportional to latency. All experiments use Llama-3.1-8B-Instruct on eight consumer-grade GPUs. FlashAttention-2 is measured with StaticCache for better efficiency, while other baselines follow their official implementations. BinaryPC achieves competitive throughput while retaining the full KV cache.

## B.7. LongBench Evaluation Details

Table 9 provides comprehensive per-dataset results on LongBench (Bai et al., 2024) across four models: Llama-3-8B, Llama-3.1-8B-Instruct (Grattafiori et al., 2024), Mistral-7B-Instruct-v0.3 (Jiang et al., 2023), and Qwen2.5-7B-Instruct-1M (Yang et al., 2025b). Across all four models, BinaryPC consistently achieves competitive or superior performance compared to baseline methods. These results demonstrate that BinaryPC maintains strong task-level accuracy across diverse task categories and model architectures, validating its effectiveness as a general-purpose sparse attention mechanism for medium-context scenarios.

## B.8. NIAH Evaluation Details

Figure 9 and Figure 10 present comprehensive Needle-in-a-Haystack (Kamradt, 2023) evaluations on Qwen2.5-7B-Instruct-1M (Yang et al., 2025b) and Llama-3.1-8B-Instruct (Grattafiori et al., 2024), respectively. Each heatmap visualizes retrieval accuracy as a function of needle depth (vertical axis) and context length from 8K to 128K tokens (horizontal axis). Static methods exhibit systematic failures at specific depth ranges due to their inability to adapt to query-dependent information needs. Query-aware and hashing-based methods improve upon static baselines but still show degradation at longer contexts or certain depth configurations. In contrast, BinaryPC and its offline variant achieve near-perfect retrieval across all positions and context lengths.

*Table 14.* RULER ([Hsieh et al., 2024](#)) benchmark task-level evaluation at 8K context length.

| Methods | Token | N-MK1 | N-MK2 | N-MK3 | N-MQ | N-MV | N-S1 | N-S2 | N-S3 | CWE | FWE | QA-1 | QA-2 | VT | Avg. |
|---|---|---|---|---|---|---|---|---|---|---|---|---|---|---|---|
| **Llama-3.1-8B** | Full | 99.00 | 99.00 | 100.0 | 99.50 | 100.0 | 100.0 | 100.0 | 100.0 | 96.30 | 91.33 | 61.00 | 80.17 | 99.80 | 94.32 |
| TOPK | 2% | 99.00 | 99.00 | 100.0 | 99.75 | 100.0 | 100.0 | 100.0 | 100.0 | 75.40 | 75.00 | 61.00 | 81.17 | 99.60 | 91.53 |
| PyramidKV | 2K | 99.00 | 98.00 | 2.00 | 99.25 | 100.0 | 100.0 | 100.0 | 99.00 | 56.40 | 77.33 | 62.00 | 81.50 | 99.80 | 82.64 |
| Cake | 2K | 97.00 | 98.00 | 84.00 | 99.50 | 100.0 | 100.0 | 99.00 | 97.00 | 88.80 | 85.00 | 60.00 | 81.17 | 99.80 | 91.48 |
| CompressKV | 2K | 99.00 | 98.00 | 55.00 | 99.00 | 99.75 | 100.0 | 100.0 | 100.0 | 87.70 | 83.33 | 61.00 | 81.17 | 99.80 | 89.52 |
| Quest | 2K | 99.00 | 98.00 | 93.00 | 99.00 | 99.75 | 100.0 | 100.0 | 98.00 | 90.60 | 86.00 | 59.00 | 81.17 | 97.20 | **92.36** |
| MagicPIG | Default | 98.00 | 97.00 | 93.00 | 97.75 | 95.25 | 98.00 | 95.00 | 93.00 | 94.10 | 87.00 | 61.00 | 80.50 | 97.00 | 91.28 |
| BinaryPC | 2% | 99.00 | 100.0 | 100.0 | 99.75 | 100.0 | 99.00 | 100.0 | 100.0 | 75.30 | 82.67 | 61.00 | 79.17 | 95.80 | 91.67 |
| BinaryPC w/ OPC | 2% | 100.0 | 99.00 | 94.00 | 98.50 | 99.75 | 100.0 | 100.0 | 100.0 | 70.80 | 77.00 | 60.00 | 80.17 | 98.00 | 90.56 |

## B.9. RULER Evaluation Details

Tables [14](#) to [18](#) provide the comprehensive per-task results on the RULER ([Hsieh et al., 2024](#)) benchmark for context lengths ranging from 8K to 128K tokens. BinaryPC maintains the most stable performance across different context lengths and subtasks, consistently achieving competitive results with full attention compared to other sparse methods.

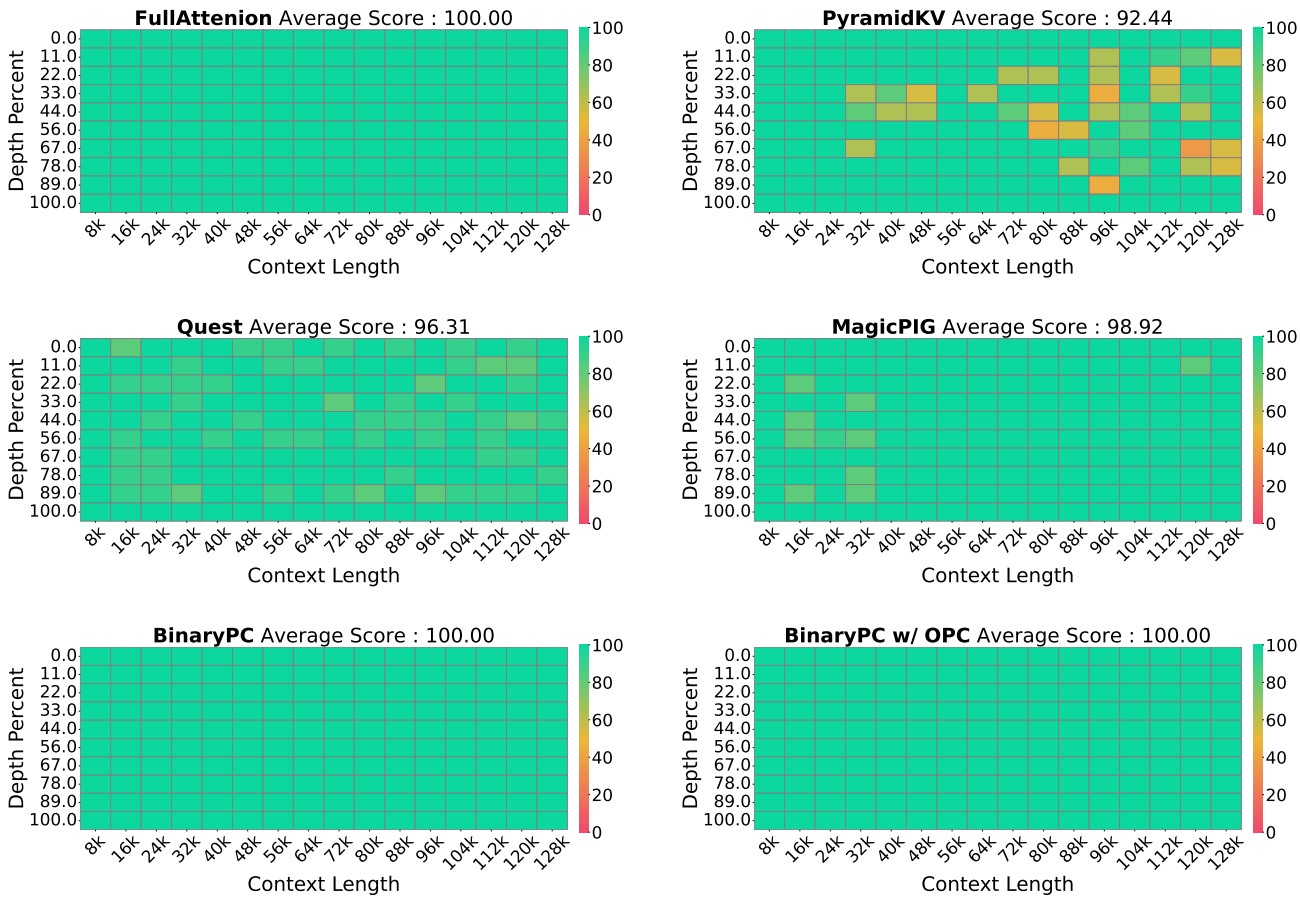

*Figure 9.* NIAH ([Kamradt, 2023](#)) evaluation on Qwen2.5-7B-Instruct-1M. PyramidKV and Quest use a 2K token budget; MagicPIG uses default settings; BinaryPC and its offline-calibrated variant use a 2% token budget.

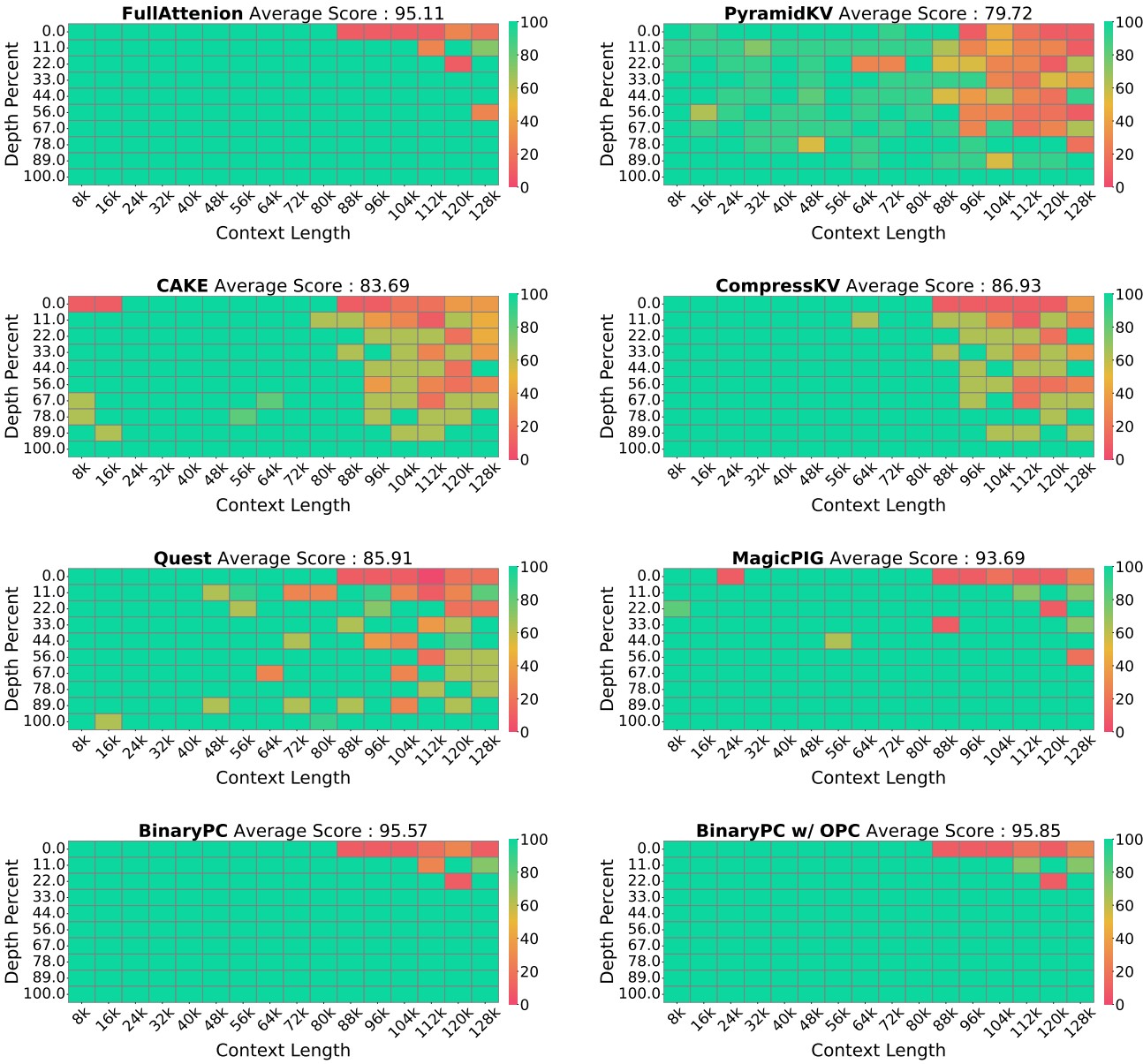

*Figure 10.* NIAH (Kamradt, 2023) evaluation on Llama-3.1-8B-Instruct. PyramidKV, Cake, CompressKV, and Quest use a 2K token budget; MagicPIG uses default settings; BinaryPC and its offline-calibrated variant use a 2% token budget.

*Table 15.* RULER (Hsieh et al., 2024) benchmark task-level evaluation at 16K context length.

| Methods | Token | N-MK1 | N-MK2 | N-MK3 | N-MQ | N-MV | N-S1 | N-S2 | N-S3 | CWE | FWE | QA-1 | QA-2 | VT | Avg. |
|---|---|---|---|---|---|---|---|---|---|---|---|---|---|---|---|
| **Llama-3.1-8B** | Full | 100.0 | 100.0 | 99.00 | 99.75 | 100.0 | 100.0 | 100.0 | 100.0 | 89.80 | 94.67 | 58.00 | 80.50 | 99.80 | 93.96 |
| TOPK | 2% | 100.0 | 100.0 | 99.00 | 99.00 | 99.50 | 100.0 | 100.0 | 100.0 | 81.50 | 73.67 | 56.00 | 79.50 | 99.60 | **91.37** |
| PyramidKV | 2K | 100.0 | 97.00 | 0.00 | 99.50 | 99.00 | 100.0 | 100.0 | 84.00 | 29.00 | 88.00 | 55.00 | 80.83 | 99.80 | 79.39 |
| Cake | 2K | 100.0 | 100.0 | 26.00 | 98.50 | 98.50 | 100.0 | 99.00 | 95.00 | 66.90 | 91.33 | 55.00 | 80.17 | 99.40 | 85.37 |
| CompressKV | 2K | 100.0 | 100.0 | 14.00 | 92.00 | 97.50 | 100.0 | 99.00 | 99.00 | 62.70 | 92.67 | 54.00 | 80.50 | 98.40 | 83.83 |
| Quest | 2K | 100.0 | 97.00 | 89.00 | 97.00 | 99.00 | 100.0 | 100.0 | 99.00 | 74.20 | 92.00 | 55.00 | 80.50 | 94.00 | 90.52 |
| MagicPIG | Default | 100.0 | 98.00 | 94.00 | 94.75 | 97.00 | 99.00 | 100.0 | 93.00 | 84.20 | 89.33 | 55.00 | 79.83 | 96.60 | 90.82 |
| BinaryPC | 2% | 100.0 | 100.0 | 98.00 | 99.50 | 99.50 | 96.00 | 100.0 | 100.0 | 75.30 | 84.00 | 56.00 | 80.50 | 92.40 | 90.86 |
| BinaryPC w/ OPC | 2% | 100.0 | 100.0 | 93.00 | 99.25 | 99.50 | 100.0 | 100.0 | 98.00 | 72.40 | 87.00 | 57.00 | 79.17 | 99.60 | 91.15 |

*Table 16.* RULER (Hsieh et al., 2024) benchmark task-level evaluation at 32K context length.

| Methods | Token | N-MK1 | N-MK2 | N-MK3 | N-MQ | N-MV | N-S1 | N-S2 | N-S3 | CWE | FWE | QA-1 | QA-2 | VT | Avg. |
|---|---|---|---|---|---|---|---|---|---|---|---|---|---|---|---|
| **Llama-3.1-8B** | Full | 100.0 | 99.00 | 97.00 | 99.00 | 100.0 | 100.0 | 100.0 | 100.0 | 12.50 | 93.00 | 55.00 | 77.17 | 99.60 | 87.10 |
| TOPK | 2% | 100.0 | 100.0 | 97.00 | 98.00 | 98.25 | 100.0 | 100.0 | 100.0 | 56.20 | 74.00 | 55.00 | 77.17 | 98.80 | **88.80** |
| PyramidKV | 2K | 99.00 | 90.00 | 0.00 | 94.75 | 95.00 | 100.0 | 100.0 | 57.00 | 3.10 | 76.67 | 53.00 | 77.50 | 99.00 | 72.69 |
| Cake | 2K | 100.0 | 99.00 | 6.00 | 98.25 | 98.25 | 100.0 | 100.0 | 89.00 | 15.20 | 80.67 | 55.00 | 76.50 | 98.00 | 78.14 |
| CompressKV | 2K | 100.0 | 98.00 | 2.00 | 96.00 | 96.75 | 100.0 | 100.0 | 97.00 | 25.10 | 82.00 | 55.00 | 76.50 | 96.40 | 78.83 |
| Quest | 2K | 99.00 | 99.00 | 63.00 | 98.25 | 98.25 | 100.0 | 100.0 | 95.00 | 15.60 | 84.67 | 56.00 | 77.75 | 93.80 | 83.10 |
| MagicPIG | Default | 99.00 | 96.00 | 91.00 | 98.00 | 96.25 | 100.0 | 100.0 | 98.00 | 8.50 | 93.33 | 55.00 | 75.08 | 98.20 | 85.26 |
| BinaryPC | 2% | 100.0 | 100.0 | 96.00 | 97.25 | 99.75 | 100.0 | 100.0 | 100.0 | 28.80 | 89.00 | 55.00 | 76.50 | 98.20 | 87.73 |
| BinaryPC w/ OPC | 2% | 100.0 | 99.00 | 96.00 | 97.50 | 99.00 | 100.0 | 100.0 | 100.0 | 26.50 | 85.67 | 55.00 | 78.17 | 98.80 | 87.36 |

*Table 17.* RULER (Hsieh et al., 2024) benchmark task-level evaluation at 64K context length.

| Methods | Token | N-MK1 | N-MK2 | N-MK3 | N-MQ | N-MV | N-S1 | N-S2 | N-S3 | CWE | FWE | QA-1 | QA-2 | VT | Avg. |
|---|---|---|---|---|---|---|---|---|---|---|---|---|---|---|---|
| **Llama-3.1-8B** | Full | 100.0 | 99.00 | 98.00 | 99.75 | 98.25 | 100.0 | 100.0 | 100.0 | 0.70 | 89.33 | 52.00 | 73.83 | 96.60 | 85.19 |
| TOPK | 2% | 100.0 | 99.00 | 98.00 | 99.75 | 98.00 | 100.0 | 100.0 | 100.0 | 4.20 | 66.00 | 52.00 | 74.50 | 92.60 | 83.39 |
| PyramidKV | 2K | 97.00 | 78.00 | 0.00 | 86.75 | 85.00 | 100.0 | 99.00 | 51.00 | 0.00 | 70.00 | 51.00 | 75.83 | 94.80 | 68.34 |
| Cake | 2K | 100.0 | 96.00 | 2.00 | 93.25 | 88.25 | 100.0 | 98.00 | 76.00 | 0.10 | 70.67 | 50.00 | 73.83 | 91.60 | 72.28 |
| CompressKV | 2K | 100.0 | 97.00 | 0.00 | 95.00 | 88.75 | 100.0 | 100.0 | 95.00 | 2.40 | 71.00 | 54.00 | 74.50 | 92.20 | 74.60 |
| Quest | 2K | 100.0 | 98.00 | 29.00 | 98.75 | 96.50 | 100.0 | 99.00 | 92.00 | 0.20 | 90.00 | 53.00 | 74.75 | 88.60 | 78.45 |
| MagicPIG | Default | 100.0 | 96.00 | 90.00 | 99.00 | 96.50 | 100.0 | 100.0 | 97.00 | 1.30 | 89.00 | 50.00 | 77.50 | 95.20 | 83.96 |
| BinaryPC | 2% | 100.0 | 100.0 | 95.00 | 98.50 | 98.00 | 100.0 | 100.0 | 99.00 | 1.70 | 84.33 | 53.00 | 74.50 | 92.60 | **84.36** |
| BinaryPC w/ OPC | 2% | 100.0 | 100.0 | 91.00 | 98.75 | 98.00 | 100.0 | 100.0 | 100.0 | 2.90 | 79.33 | 54.00 | 75.83 | 94.00 | 84.14 |

*Table 18.* RULER (Hsieh et al., 2024) benchmark task-level evaluation at 128K context length.

| Methods | Token | N-MK1 | N-MK2 | N-MK3 | N-MQ | N-MV | N-S1 | N-S2 | N-S3 | CWE | FWE | QA-1 | QA-2 | VT | Avg. |
|---|---|---|---|---|---|---|---|---|---|---|---|---|---|---|---|
| **Llama-3.1-8B** | Full | 97.00 | 83.00 | 70.00 | 99.00 | 94.00 | 100.0 | 100.0 | 100.0 | 0.10 | 74.67 | 42.00 | 72.58 | 62.00 | 76.49 |
| TOPK | 2% | 97.00 | 82.00 | 59.00 | 99.00 | 92.50 | 100.0 | 100.0 | 100.0 | 0.10 | 55.33 | 42.00 | 72.58 | 57.60 | **73.62** |
| PyramidKV | 2K | 96.00 | 28.00 | 0.00 | 42.75 | 33.25 | 100.0 | 98.00 | 13.00 | 0.20 | 60.33 | 42.00 | 69.17 | 48.00 | 48.52 |
| Cake | 2K | 95.00 | 73.00 | 0.00 | 90.75 | 75.00 | 100.0 | 100.0 | 49.00 | 0.20 | 40.33 | 42.00 | 72.58 | 62.00 | 61.61 |
| CompressKV | 2K | 96.00 | 73.00 | 0.00 | 93.50 | 80.25 | 100.0 | 100.0 | 75.00 | 0.20 | 51.67 | 42.00 | 72.50 | 65.00 | 65.32 |
| Quest | 2K | 95.00 | 66.00 | 0.00 | 92.00 | 85.25 | 100.0 | 100.0 | 53.00 | 0.20 | 63.00 | 43.00 | 65.83 | 63.40 | 63.59 |
| MagicPIG | Default | 96.00 | 75.00 | 42.00 | 96.25 | 86.00 | 100.0 | 98.00 | 94.00 | 0.10 | 75.00 | 42.00 | 70.92 | 61.60 | 72.07 |
| BinaryPC | 2% | 97.00 | 80.00 | 45.00 | 98.75 | 92.50 | 100.0 | 100.0 | 100.0 | 0.10 | 71.00 | 43.00 | 73.58 | 51.00 | 73.23 |
| BinaryPC w/ OPC | 2% | 96.00 | 79.00 | 44.00 | 97.75 | 92.50 | 100.0 | 100.0 | 100.0 | 0.10 | 71.33 | 43.00 | 72.58 | 56.20 | 73.27 |

