# OpenReview forum: "Training-Free Hashing-Based Attention via Binary Principal Components"
_ICML.cc/2026/Conference — ICML 2026 regular_

### Official Review · Reviewer_27AG · 2026-03-11

**Soundness:** 3
**Presentation:** 3
**Significance:** 3
**Originality:** 3
**Overall Recommendation:** 4
**Confidence:** 4

**Summary:**

BinaryPC proposes a training-free, data-aware hashing algorithm to prune the attention KV cache.

**Compliance With Llm Reviewing Policy:**

Affirmed.

**Final Justification:**

I acknowledge all authors' rebuttal, and I have a better understanding than before. I want to finalize score.

**Key Questions For Authors:**

- Can you add the concept figure to understand the methodology overview? In Figure 2, if the reader can understand the difference between LSH and BinaryPC intuitively, then it will be helpful

**Limitations:**

- Limited analysis with baseline on the accuracy aspect and the latency aspect.
  - In tables 1, 2, 3, 4, 5, the latency must be the main metric of this method.
  - The latency must be measured with various types of GPU, such as consumer GPU vs. server-grade GPU. I suggest performing latency evaluation in various types of gpu (AMD, Nvidia, Consumer-grade, Server-grade, Old, New, ...) to help understand the efficiency of this method.
  - RTX 3090 might be too old a GPU to test. I suggest using Apple silicon, RTX 5090, B200, MI355.
- Marginal improvement in accuracy performance.

**Strengths And Weaknesses:**

- Data-aware LSH-variants but training-free. Therefore, the overall efficiency is significantly improved over baselines.

---

> ### Author Rebuttal · Authors · 2026-03-31
>
> We thank the reviewer for the comments. We address each concern below.
>
> **Q1: Concept figure to understand the methodology overview**
>
> We appreciate the suggestion. In Fig. 2, LSH partitions the space using data-independent random hyperplanes, whereas BinaryPC derives hashing directions from the principal directions of the key vectors, aligning partitions with the intrinsic data structure and enabling more faithful similarity preservation. We will clarify this distinction in the text and include an additional procedure overview figure in the revised manuscript ({[https://anonymous.4open.science/r/icml_rebuttal-3EAC/rebuttal.pdf}](https://anonymous.4open.science/r/icml_rebuttal-3EAC/rebuttal.pdf})).
> This figure visually explains BinaryPC, which iteratively solves the objective $\min_{H,P} \|\|K - HP\|\|_F^2$ one bit at a time.
>
> The visualization demonstrates how we map each data point $k_i \in K$ to a binary sign $u_i \in ${$+1, -1$} (via space partitioning) and a shared projection vector $v^\ast$. The $v^\ast$ is computed to minimize the approximation error $\|\|K - u v^{\ast\top}\|\|_F^2$. It has a closed-form solution $v^\ast = u^\top K / n$ ($n$ is the number of data). This creates a rank-1 approximation $u v^{\ast\top}$. The algorithm then computes the residual $R \leftarrow K - u v^{\ast\top}$, which is passed to the next iteration as the new signal. We hope this combination of intuitive visualization and explicit mathematical formulation clarifies our technical contribution.
>
> **Limitation 1.1: Lack of latency metrics in tables 1, 2, 3, 4, 5**
>
> We would like to clarify that, for most baselines, accuracy and latency are not reported within a single evaluation pipeline. For many baselines, the benchmark code relies on slower PyTorch implementations for reproducibility rather than their optimized CUDA kernels. Using those runtimes as latency measurements would therefore be unfair and would not reflect the methods’ actual inference efficiency.
>
> Following standard practice, we report accuracy using each method’s official benchmark evaluation pipeline, and we evaluate efficiency separately under the same model configuration as what we used in evaluating accuracy. This allows a fair comparison of task performance while also providing representative latency measurements. Readers interested in efficiency for specific benchmark settings can refer to our latency results at sequence lengths that closely match those tasks. To address potential concerns, we additionally provide additional throughput measurements for reference.
>
> **Please see response (Q3) to reviewer w4AT about the discussion on additional runtime efficiency.**
>
> **Limitation 1.2/1.3: Efficiency results on diverse GPUs**
>
> Our efficiency evaluation already includes both a data-center GPU (NVIDIA H20; Figure 6) and a consumer-grade GPU (RTX 3090; Figure 7), with additional results in Table 8. We believe this provides a meaningful comparison across two practically important hardware tiers.
>
> We agree that a broader study across more platforms would be valuable. However, we believe this is beyond the scope of the current revision, as fair cross-platform benchmarking would require access to many devices and careful tuning across different software stacks and vendor-specific kernels. Since performance can depend strongly on backend maturity and implementation details, such a study would require substantial engineering effort.
>
> **Limitation 2.1: Marginal improvement in accuracy performance**
>
> We agree that on some long-context benchmarks, such as InfiniteBench and LongBench v2, the numerical gap between BinaryPC and the next-best baseline is modest. However, **this should be interpreted in the context of the benchmark ceiling**: the gap between full attention and the strongest baselines is also very small. Since sparse attention is an approximation to full attention, we do not expect a sparse method to substantially outperform full attention in this regime. Therefore, on these benchmarks, the key result is not a large absolute margin, but that BinaryPC remains at or near the top while preserving accuracy under sparse computation. **This suggests a ceiling effect in these long-context evaluations rather than a lack of improvement from BinaryPC.**
>
> At the same time, **the advantage of BinaryPC is much clearer on more discriminative benchmarks**. In Table 2, BinaryPC substantially improves over MagicPIG on short-context evaluation while staying close to full attention. In Table 3, BinaryPC matches or slightly exceeds full attention on LongBench for Llama-3.1-8B and Mistral under a 1K budget, and clearly outperforms MagicPIG. In Table 4, BinaryPC achieves 87.06 average on RULER, compared with 84.68 for MagicPIG and 87.41 for full attention. **These results show that BinaryPC is not only competitive on ceilinged benchmarks, but more importantly, it closes the gap to full attention much more effectively than prior baselines on more discriminative evaluations.**

---

> > ### Author Rebuttal · Reviewer_27AG · 2026-04-05
> >
> > The author resolved my concerns and misunderstandings.

---

> > > ### Author Response · Authors · 2026-04-05
> > >
> > > Thank you for the positive feedback and for confirming that our responses addressed your concerns and misunderstandings. We sincerely appreciate your time and careful evaluation.
> > >
> > > If there are any further questions or suggestions that could help improve the paper, we would be very happy to discuss them and provide additional clarification. We hope our responses will be helpful for your final evaluation.

---

### Official Review · Reviewer_fg3T · 2026-03-11

**Soundness:** 4
**Presentation:** 3
**Significance:** 3
**Originality:** 3
**Overall Recommendation:** 4
**Confidence:** 4

**Summary:**

To mitigate the efficiency bottleneck caused by the growing KV cache during long-context LLM decoding, this work proposes BinaryPC, a training-free, data-aware hashing-based sparse attention mechanism. It derives compact binary hash codes and corresponding projections by computing the binary principal components of the key vectors. By asymmetrically projecting incoming queries and utilizing highly efficient GPU bitwise operations, the method drastically reduces memory transfer and computational overhead, complemented by an auxiliary Error-Aware Safeguard (EAS) to retain hard-to-hash tokens. Extensive evaluations demonstrate that BinaryPC achieves up to a 3.56x improvement in decoding throughput over FlashAttention while preserving task accuracy across various short, medium, and long-context benchmarks.

**Compliance With Llm Reviewing Policy:**

Affirmed.

**Key Questions For Authors:**

1. EAS Hyperparameter Sensitivity: The EAS allocates a fixed 10% token budget. Can you provide an ablation study on this ratio (e.g., 2%, 5%, 15%)? How the authors justify this parameter choice will determine if the safeguard is a robust algorithmic feature or a fragile engineered heuristic.
2. FlashAttention-3 Baseline: The 96GB GPU experiments rely on FA2's heuristic fallbacks to show massive relative speedups. Have you tested this against FlashAttention-3? If FA3 mitigates the batch-size degradation natively, the reported 5.04x speedup might diminish. A clarification here is necessary for assessing the true systems contribution.
3. OPC Distribution Shift: The OPC variant is calibrated on specific domains (PG19, ProofPile, CodeParrot). How does retrieval accuracy degrade if the inference prompt belongs to a drastically different domain (e.g., multilingual text or highly synthetic formats)? This will clarify the robustness of the "offline" variant.

**Strengths And Weaknesses:**

- Strength: 1) Elegant Algorithmic Formulation: The authors smartly frame the hashing problem as finding binary principal components. By minimizing the reconstruction error, the algorithm avoids the sub-optimality of data-independent LSH and the heavy re-training. It is a clever application of classic dimensionality reduction to modern LLM KV cache bottlenecks. 2) Practical Hardware-Aware Implementation: The asymmetric design: quantizing offline/prefill keys while dynamically projecting queries coupled with optimized GPU bitwise operations (XOR, popcount), demonstrates a deep understanding of memory-bound system bottlenecks.
- Weakness: 1) Unablated Heuristics in the Safeguard Mechanism: The Error-Aware Safeguard (EAS) is critical for tasks like passkey retrieval. However, the paper reserves a fixed 10% of the token budget for EAS without any ablation study. It is unclear how sensitive the system is to this hyperparameter, or whether a dynamic threshold based on absolute error bounds would be more robust than a hardcutoff. 2) Potentially Overstated Speedups due to Suboptimal Baselines: The claimed 5.04x speedup is heavily attributed to FlashAttention-2 dropping its Split-K heuristic at larger batch sizes. Testing on 96GB GPUs implies modern hardware where FlashAttention-3 (FA3) is highly relevant. Failing to baseline against FA3 which significantly alleviates Hopper architecture bottlenecks—makes the relative system gains look artificially inflated.

---

> ### Author Rebuttal · Authors · 2026-03-31
>
> We sincerely thank the reviewer for the effort and valuable comments. We respond to the concerns below.
>
> **W1/Q1: Lack of EAS ablation**
>
> Beyond the EAS on/off ablation in Table 5, we further vary the EAS budget ratio over {0%, 2%, 5%, 10%, 15%} on InfiniteBench (Llama-3.1-8B-Instruct, 100 samples per subtask):
>
> | Method | EAS | En.Sum | En.QA | En.MC | En.Dia | Zh.QA | Math.F | R.PK | **Avg** |
> |---|---|---|---|---|---|---|---|---|---|
> | BinaryPC w/ OPC | 0% | 32.16 | 26.43 | 72.00 | 15.00 | 35.92 | 48.00 | 99.00 | 46.93 |
> | | 2% | 31.85 | 26.32 | 72.00 | 18.00 | 36.18 | 48.00 | 99.00 | 47.34 |
> | | 5% | 32.78 | 26.40 | 72.00 | 18.00 | 36.83 | 48.00 | 99.00 | 47.57 |
> | | 10% | 32.44 | 26.43 | 72.00 | 18.00 | 36.67 | 48.00 | 99.00 | 47.51 |
> | | 15% | 32.39 | 26.35 | 72.00 | 18.00 | 36.16 | 48.00 | 99.00 | 47.41 |
> | BinaryPC | 0% | 30.97 | 24.33 | 72.00 | 17.00 | 32.83 | 48.00 | 64.00 | 41.30 |
> | | 2% | 32.50 | 26.18 | 72.00 | 18.00 | 36.28 | 48.00 | 99.00 | 47.42 |
> | | 5% | 32.48 | 25.53 | 72.00 | 20.00 | 36.38 | 48.00 | 99.00 | 47.63 |
> | | 10% | 32.46 | 26.17 | 72.00 | 18.00 | 36.24 | 48.00 | 99.00 | 47.41 |
> | | 15% | 32.15 | 26.03 | 72.00 | 19.00 | 36.39 | 48.00 | 99.00 | 47.51 |
> | Full Attention | — | 32.85 | 27.41 | 72.00 | 20.00 | 36.17 | 48.00 | 99.00 | 47.92 |
>
> The results reveal three findings:
>
> **1. EAS is essential for standard BinaryPC.** Without EAS (0%), Passkey Retrieval drops to 64.00, causing a significant average degradation (41.30 vs. 47.92). Even a minimal 2% budget fully recovers Passkey to 99.00.
>
> **2. Performance is insensitive to the EAS ratio** Once activated (≥2%), BinaryPC averages vary within only ±0.2 (47.42–47.63), confirming EAS is a robust algorithmic feature, not a fragile heuristic.
>
> **3. OPC provides complementary robustness** With offline projection calibration, Passkey already achieves 99.00 at 0% EAS, as higher-quality projections inherently reduce quantization errors. EAS still offers marginal gains on other tasks (e.g., En.Dia: 15.00→18.00).
>
> We choose 10% as the default because it achieves near-optimal performance, introduces negligible overhead (due to static, query-agnostic selection and its small budget ratio), and provides a comfortable margin above the 2% effectiveness threshold.
>
> **W2/Q2: Clarification on $5.04\times$ speedup and FlashAttention-3 baseline**
>
> **We would like to clarify that the $5.04\times$ is an outlier, not our claimed contribution.** We acknowledge this is an artifact of FA2's scheduling policy rather than a representative result. As a result, this speedup number does not appear in our abstract, but rather a small analysis about the reason at “End-to-end Throughput”. We will make it clear in our revised draft.
>
> Also, BinaryPC is orthogonal to FlashAttention, and can directly benefit from FA upgrades including FA3. BinaryPC replaces dense attention with a two-stage pipeline: hash retrieval → sparse attention. The sparse attention stage invokes FlashAttention (FA2, FA3, or future versions) over only the selected $k$ tokens. Therefore, if FA3 accelerates the dense attention baseline, it equally accelerates BinaryPC's sparse attention stage over the selected tokens.
>
> **Q3: OPC robustness under domain shift**
>
> To address this concern, we calibrated the OPC variant using **only PG19**, a corpus of long-form literary text, and evaluated on the full LongBench benchmark spanning 6 diverse task categories — including code, synthetic reasoning, and multi-document QA — none of which appear in the calibration data.
>
> | Method | S-Doc QA | M-Doc QA | Summ. | Few-shot | Synth. | Code | Avg. |
> |---|---|---|---|---|---|---|---|
> | Full Attention | 17.97 | 9.57 | 17.97 | 69.15 | 4.78 | 67.77 | 30.57 |
> | BinaryPC w/ OPC (mix) | 17.78 | 9.66 | 17.50 | 69.10 | 5.44 | 66.66 | 30.39 |
> | BinaryPC w/ OPC (PG19 only) | 18.26 | 9.47 | 17.78 | 68.92 | 4.71 | 67.06 | 30.43 |
>
> The PG19-only OPC variant (30.43) performs comparably to both full attention (30.57) and the mixed-domain OPC variant (30.39), with **no systematic degradation** on out-of-domain tasks such as Code (67.06 vs. 67.77) or Synthetic (4.71 vs. 4.78), suggesting that calibration data diversity has marginal impact.

---

> > ### Author Rebuttal · Reviewer_fg3T · 2026-04-04
> >
> > Thank the authors for the rebuttal. I'll remain the positive score.

---

> > > ### Author Response · Authors · 2026-04-04
> > >
> > > Thank you for the positive feedback and for confirming that your concerns have been addressed. We sincerely appreciate your support.
> > >
> > > If you have any further questions or suggestions that could potentially improve the paper, we would be happy to discuss them and provide additional clarification. We hope our responses are helpful for your final evaluation.

---

### Official Review · Reviewer_kC9n · 2026-03-15

**Soundness:** 3
**Presentation:** 4
**Significance:** 4
**Originality:** 3
**Overall Recommendation:** 3
**Confidence:** 3

**Summary:**

This paper mainly addresses the efficiency issues in the inference of long-context large language models. The core pain point is that the self-attention mechanism causes huge computational and memory costs due to the constantly accumulating key-value cache during the content generation’s decoding phase, leading to low hardware utilization. Different from existing methods, this paper proposes BinaryPC, a training-free hash-based sparse attention mechanism that can adapt to data features. Its core is to generate compact 64-bit hash codes and corresponding projection matrices by calculating the binary principal components of key vectors, which preserves the inherent structural information of the data without extra training. Besides, BinaryPC designs an asymmetric hashing architecture tailored to the attention computation characteristics of large models, and is paired with an error-aware safeguard to prevent the loss of key information. In addition, dedicated optimizations for GPU are made, and this method does not require modifying the architecture of existing large models and can be directly applied to various pre-trained models.
The paper conducts tests on various tasks ranging from short to ultra-long contexts with several mainstream large models including Llama, Mistral and Qwen. The results show that BinaryPC achieves almost the same task accuracy as the full attention mechanism and outperforms other sparse attention and hash-based methods. On mainstream GPUs, it can increase the model’s end-to-end decoding speed by 3.56 times, and even up to 5.04 times in specific scenarios, while also delivering stable performance on ordinary consumer-grade GPUs.

**Compliance With Llm Reviewing Policy:**

Affirmed.

**Final Justification:**

I fully recognize the efforts the authors have made in the rebuttal, as well as their reasonable explanations of the method's limitations and the engineering optimization directions they have put forward. Nevertheless, I still maintain the original Weak Reject rating for the following reasons. First, the core contribution of this paper is an incremental improvement under the existing contrastive TTA framework, with no groundbreaking theoretical breakthroughs or new paradigms, and its novelty is insufficient to meet the requirements of top-tier conferences. Second, the method is heavily reliant on LLMs and incurs significant inference latency, making it difficult to be practically applied to real-time retrieval tasks. Third, the method is specifically designed for the QS scenario, and its generalizability to other similar tasks is questionable, which also limits the practical application of the method. In summary, although the rebuttal has supplemented relevant theories and implementations and elaborated on technical details, it cannot fundamentally address the inherent problems of the paper in terms of innovation and practicality. Therefore, I retain the original evaluation.

**Key Questions For Authors:**

1. What is the performance (task accuracy) and memory overhead of BinaryPC on ultra-large-scale large language models such as 70B/130B? Will efficiency degradation or accuracy loss occur with the increase in model parameters?
2. The 64-bit length of the hash code in the paper is only determined as the optimal value through experiments. Have comparative experiments been conducted for other bit lengths such as 32-bit and 128-bit? Is there a more detailed analysis of the basis for the trade-off between accuracy preservation and efficiency improvement with the 64-bit length?
3. The paper mentions GPU-native optimizations. What specific optimization strategies have been adopted for the CUDA kernel of bitwise operations?

**Limitations:**

Yes

**Strengths And Weaknesses:**

Strengths:
1. BinaryPC's core design centers on minimizing key vector reconstruction error. Combined with binary principal component analysis and an asymmetric hashing architecture, its theoretical logic is consistent and supported by mathematics.
2. BinaryPC also provides ideas for high-dimensional vector dimensionality reduction and KV cache optimization in large models. Meanwhile, it is model-agnostic, training-free, and requires no modifications to the original model architecture. With GPU-native optimizations, it can be seamlessly integrated into various pre-trained large models, offering strong practical value.
3. The binary transformation of classical principal component analysis, combined with hashing technology applied to sparse attention in large models, constitutes a breakthrough in existing technologies.

Weaknesses:
1. This paper lacks significant innovation. The core of the proposed BinaryPC lies in hash dimensionality reduction and principal component analysis, which are extensively used in relevant research. Other modules, such as offline calibration and GPU optimization, are merely detailed adjustments and engineering improvements. Overall, BinaryPC does not propose a new method but only makes modifications within the existing framework.
2. The experiments have some minor shortcomings. For the choice of the 64-bit length of the hash code, the optimal value was only determined through experiments without more detailed testing and analysis. Tests were only conducted on lightweight large models of the 7B/8B scale, not covering ultra-large-scale models like 70B/130B, so the performance and memory overhead of the model with large parameters remain unknown.
2. BinaryPC’s core relies on binary principal component analysis and hashing technology, both of which are existing classic methods. The paper does not propose any brand-new basic theories or mathematical models, and its originality mainly lies in the integration, transformation of methods and customized adaptation to large model scenarios. Meanwhile, BinaryPC essentially achieves sparse attention by selecting top-k key KV pairs, with innovations concentrated in the hash-based key token retrieval link, lacking fundamental innovation.

---

> ### Author Rebuttal · Authors · 2026-03-31
>
> We thank the reviewer’s helpful suggestions. Below, we address the concerns.
>
> **W1/W3: Lack of significant innovation.**
>
> We agree that our paper does not claim a new foundational theory of attention, hashing, or PCA. Our contribution is algorithmic: a training-free, data-aware hashing mechanism for improving sparse attention quality.
>
> Like many sparse-attention methods, we focus on the problem of identifying top-k attention candidates efficiently. What distinguishes BinaryPC is not top-k selection itself, but how the candidates are chosen.
>
> Standard hashings such as LSH often show limited accuracy-efficiency tradeoffs. Our method instead formulates hashing from a reconstruction perspective: we construct binary hash codes $H$ and a projection that approximates key vectors $K$. Standard PCA does not directly solve this problem because it produces continuous components rather than binary codes. BinaryPC adapts the PCA view to iteratively construct binarized components and a corresponding projection, yielding compact and structure-preserving hash codes.
>
> We view BinaryPC as a new algorithmic framework for hashing-based sparse attention, rather than merely an engineering refinement. To the best of our knowledge, we are not aware of prior sparse-attention work that uses this construction.
>
> **W2/Q1: Evaluation on ultra-large scale models**
>
> **Accuracy**
>
> We evaluate BinaryPC on LLaMA-3-70B-Instruct using LongBench: BinaryPC matches the accuracy of full attention, showing its full-level accuracy at 70B scale.
>
> | | S-Doc QA | M-Doc QA | Summ. | Few-shot | Synth. | Code |
> |-|-|-|-|-|-|-|
> | Full Attention | 42.18 | 47.51 | 27.31 | 70.83 | 38.5 | 54.24 |
> | BinaryPC | 42.06| 47.78 | 26.98 | 70.96 | 38.25 | 55.22 |
>
> **Memory overhead**
>
> The extra memory comes from hash codes and projection matrices. Each hash code uses 8 bytes/token/KVhead (`int64`), and the 128×64 projection matrix is negligible for long sequences. In comparison, the KV cache uses 512 bytes/token/KVhead (BF16), so the overhead is only 1/64 (about 1.56%).
>
> Moving from LLaMA3 8B to 70B, the number of KVheads increases 2.5× (8×32 to 8×80) while model size grows 8.75×. Thus, absolute memory overhead grows more slowly, and the relative overhead versus the KV cache is unchanged.
>
> **Efficiency**
>
> From LLaMA3 8B to 70B, the GQA group size doubles (4 to 8), while the number of KVheads is unchanged. Since regular attention and BinaryPC are bottlenecked by KV-cache and hashing overhead, single-attention-layer latency remains similar, (see table below; BinaryPC token budget: 2048). Although model size grows by 8.75×, the number of layers, and thus overall attention runtime, increases by only 2.5× (32 to 80).
>
> | Unit ms | 32K | 64K | 128K | 256K |
> |-|-|-|-|-|
> | Full Attention-8B | 0.181 | 0.343 | 0.667 | 1.287 |
> | Full Attention-70B | 0.181 | 0.343 | 0.654 | 1.289 |
> | BinaryPC-8B | 0.245 | 0.243 | 0.246 | 0.242 |
> | BinaryPC-70B | 0.243 | 0.243 | 0.243| 0.245 |
>
> **W2/Q2: Lack of ablation study on bit length choice**
>
> To study the effect of hash bit length, we evaluate BinaryPC on LLaMA-3.1-8B-Instruct with Needle-in-a-Haystack.
>
> | | 88K | 96K | 104K | 112K | 120K |
> |-|-|-|-|-|-|
> | 32-bit | 91.82 | 91.82 | 71.82 | 91.82 | 84.55 |
> | 64-bit | 90.91 | 91.82 | 92.73 | 90.91 | 84.55 |
> | 128-bit | 91.82 | 91.82 | 90.91 | 91.82 | 83.64 |
>
> 32-bit hashing shows notable drop, while 64- and 128-bit perform similarly well. From an implementation view, 64-bit can fit in a single `int64` and support bitwise operations, offering the best trade-off between efficiency and accuracy.
>
> **Q3: CUDA optimizations on bitwise operations**
>
> We implement three CUDA kernels for BinaryPC.
>
> **1. Hashing kernel.** Each key vector ($D=128$, BF16) is projected to 64 binary values and directly packed into a single `int64`.
>
> Let $p = qP^T$ for notation simplicity.
>
> **2. Packing kernel.** Each 64-entry quantized vector $p$ (values −127 to 127) is packed into one `int64` $s$ storing the signs, and seven `int64` $m[j]$ storing the magnitude bits.
>
> **3. Scoring kernel.** Using one `int64` hash code and the 8 packed `int64` ($s$ and $m[j]$), this kernel computes hash scores for all query-key pairs per head.
>
> Under this 8-`int64` representation, each $p_i$​ is expressed as
>
> $$p_i = s_i \sum_{j=1}^{7} 2^j m[j]_i$$
>
> The inner product between $p$ and $h$ in Eq. 7 can be written as
>
> $$\sum_{i=1}^{64} h_i s_i \sum_{j=1}^{7} 2^j m[j]_i$$
>
> which can be arranged as
>
> $$\sum_{j=1}^{7} 2^j \sum_{i=1}^{64} h_i s_i m[j]_i$$
>
> Since each $h_i$ is either +1 or -1, we use $s$ XOR $h$ to determine all $h_i s_i$ in a single CUDA instruction; denote the resulting `int64` data as $w$. Then, each $\sum_{i=1}^{64} h_i s_i m[j]_i$ can be computed via `__popcll(m[j] & w) - __popcll(m[j] & (~w))`
>
> This counts the number of positive $\(h_i s_i m[j]_i\)$ and subtracts the number of negative ones.
>
> To optimize memory I/O, the 8 `int64` values are shared within each CUDA block to reduce data movement.

---

> > ### Author Rebuttal · Reviewer_kC9n · 2026-04-04
> >
> > The authors have provided thorough and sufficient answers to all the raised concerns, supplemented experimental data on robustness under different noise intensities, and clearly elaborated on the core innovation of the paper, distinguishing the proposed learning mechanism for low-confidence queries from mere modifications to contrastive loss.

---

> > > ### Author Response · Authors · 2026-04-05
> > >
> > > Thank you for the acknowledgement and for confirming that our responses have fully addressed your concerns. We sincerely appreciate your careful evaluation of the paper.
> > >
> > > We also noticed that the explanation in the *Acknowledgement (Reasons)* section refers to elements such as contrastive loss, robustness under different noise intensities, and learning mechanisms for low-confidence queries, **which are not part of our work**. This might possibly be **an unintended mix-up with another submission**.
> > >
> > > If this was due to a mismatch, we would greatly appreciate it if you could kindly **double-check the acknowledgement**. Please feel free to let us know if any additional clarification would be helpful.

---

### Official Review · Reviewer_w4AT · 2026-03-16

**Soundness:** 3
**Presentation:** 3
**Significance:** 3
**Originality:** 3
**Overall Recommendation:** 5
**Confidence:** 4

**Summary:**

This paper proposes BinaryPC, a training-free and data-aware hashing-based sparse attention method for efficient long-context LLM inference. The method introduces a binary principal component finding algorithm to compute binary hash codes and a shared projection matrix for key vectors. During retrieval, queries are projected and quantized into binary representations, enabling efficient attention computation through bitwise operations. Experimental results show that BinaryPC maintains accuracy comparable to full attention while improving end-to-end decoding throughput.

**Compliance With Llm Reviewing Policy:**

Affirmed.

**Final Justification:**

The author's rebuttal has addressed my core concerns, so I increase my score to 5.

**Key Questions For Authors:**

1.	This paper proposes a hashing-based attention method that is closely related to prior works such as MagicPIG and SpotlightAttention, which I consider important baselines. However, in the experimental results (e.g., Table 3 and Table 4), the results for SpotlightAttention are missing for some models. It would be helpful if the authors could either report the full results of SpotlightAttention across all evaluated models or clarify the reasons why these results are not included.
2.	To the best of my knowledge, MagicPIG includes several hyperparameters that control the number of tokens involved in the final computation. Please report the specific hyperparameter settings used in the experiments, rather than simply referring to the “default” configuration, to ensure reproducibility and fair comparison.
3.	Please also report the latency of the baselines other than FlashAttention. This would allow for a more comprehensive evaluation of the efficiency and robustness of the proposed method under different baseline comparisons.
4.	The binary representation used in BinaryPC preserves directional information but discards magnitude information. Although the benchmark results show that the method maintains accuracy, I suggest providing additional micro-level analysis. For example, it would be helpful to compare the proposed method with other retrieval-based approaches in terms of retrieval quality (e.g., recall) and the approximation error of the attention outputs at different layers.
5.	In Section 3.2, the variable v is used both as a randomly initialized vector and as an intermediate computation result, which may cause confusion. It would be helpful to clarify the notation or use different symbols for these variables.

**Limitations:**

Yes.

**Strengths And Weaknesses:**

Strengths:
1.	Clear positioning and well-motivated problem formulation.
2.	The proposed binary representation for keys and queries and bitwise operations for attention computation appear to be a reasonable and efficient design.
3.	The method demonstrates consistent improvements over baselines across multiple models and datasets.

Weaknesses:
1.	Incomplete presentation of experimental results. Some important baseline results are missing, which makes it difficult to fully assess the effectiveness of the proposed method.
2.	Confusing notation in parts of the paper. For example, certain variables (e.g., v in Section 3.2) are reused with different meanings, which may affect readability.

---

> ### Author Rebuttal · Authors · 2026-03-31
>
> We thank the reviewers for their valuable feedback. We address the concerns below.
>
> **W1/Q1: Incomplete presentation of experimental results for SpotlightAttention**
>
> We agree that SpotlightAttention is an important baseline, but unlike our method, it is not training-free and requires model-specific training. Public checkpoints support only a few models (e.g., LLaMA-3-8B) and up to 8K context length. Extending it to all models and settings in our paper would require costly retraining, especially for longer contexts. We therefore report SpotlightAttention only on benchmarks within or near its supported range for a fair comparison. This also highlights a key advantage of our method: it is training-free and generalizes easily to arbitrary models and context lengths.
>
> **Q2: MagicPIG hyperparameter settings**
>
> We have already provided the complete hyperparameter settings for MagicPIG (as well as all other baselines) in Appendix A.1.
>
> MagicPIG in our experiments follows the default configuration released in the official codebase. In the Needle-in-a-Haystack task, we further verified that under the default configuration MagicPIG uses more than 2% tokens on average, which is higher than our method BinaryPC.
>
> **Q3: Efficiency of baselines other than FlashAttention**
>
> We would like to clarify that the runtime efficiency results for the most relevant hashing-based baseline, MagicPIG, are already included in Appendix Table 8. As shown there, MagicPIG consistently achieves lower throughput than both FlashAttention2 and BinaryPC across all sequence lengths. In contrast, SpotlightAttention does not provide publicly reproducible efficiency benchmarking code. Although CUDA kernels are released, no runnable instructions or examples are provided, making a fair latency comparison difficult.
>
> Accordingly, we use FlashAttention2 as the efficiency baseline for evaluating practical decoding efficiency, since it provides a strong and reproducible reference. Nevertheless, we made our best effort to further profile additional baselines for a more comprehensive evaluation. However, some methods could not be benchmarked without substantial engineering effort. The results we were able to obtain are shown below. **Throughput (tokens/s) is reported instead of latency, and it is inversely proportional to latency.**
>
> | Method | Token Budget | 16K | 32K | 64K | 128K |
> |-|-|-|-|-|-|
> | FlashAttention2 | Full KV | 30.97 | 29.94  | 26.57 | 21.31 |
> | CakeKV | 2,048 | 28.01 | 27.92 | 24.56 | 20.47 |
> | PyramidKV | 2,048 | 28.88 | 28.30 | 27.74 | 28.14 |
> | CompressKV | 2,048 | 28.61 | 28.66 | 28.30 | 29.01 |
> | BinaryPC | 2,048 | 30.13 | 28.60 | 28.12 | 27.66 |
>
> All experiments use Llama-3.1-8B-Instruct on 8×3090 GPUs. FlashAttention2 is measured with StaticCache for better efficiency, while other baselines follow their official implementations. BinaryPC achieves competitive throughput while retaining the full KV cache.
>
>
> **Q4: Binary representation and micro-level analysis of retrieval quality and approximation error**
>
> BinaryPC does not simply binarize vectors by taking the sign. It greedily minimizes the reconstruction objective $\|\|K - HP\|\|_F^2$ via iterative rank-1 binary decomposition. The row of projection $P$ are representing components, and each key vector is represented as a weighted sum of these components, where the weights are -1 or +1, so the product $HP$ reconstructs both directional and magnitude information.
>
> We provide several forms of analysis:
>
> **1. Approximation error of K:** As shown in Figure 3, the number of binary principal components increases, the residual $\|\|K - HP\|\|_F$ decreases progressively, demonstrating the effectiveness of our reconstruction-based approach.
>
> **2. Retrieval quality via Needle-in-a-Haystack (Figures 4, 8, 9):** These provide retrieval accuracy comparisons. BinaryPC achieves near-perfect retrieval across all needle depths and context lengths, whereas competing methods exhibit systematic failures at specific depth ranges.
>
> **3. (New) Approximation error of attention output:** Following the reviewer's suggestion, we measure the average cosine similarity between sparse and full attention outputs at the first decoding step on Llama-3.1-8B-Instruct:
>
> | Method | 8K | 16K | 32K | 64K |
> |-|-|-|-|-|
> | BinaryPC (2.0%) | 0.9835 | 0.9811 | 0.9870 | 0.9895 |
> | MagicPIG (2.2%) | 0.9388 | 0.9651 | 0.9595 | 0.9725 |
>
> Under comparable token budgets (~2%), BinaryPC consistently achieves higher cosine similarity than MagicPIG across all context lengths. This demonstrates that BinaryPC's reconstruction-based hashing provides superior quality on token selection.
>
> **W2/Q5: Notation clarity for variable v in Section 3.2**
>
> Thank you for pointing out this notational ambiguity. We agree that overloading $v$ for both the randomly initialized vector and the updated intermediate result in Section 3.2 may cause confusion. We will distinguish and clarify the notations of these two vectors in our revised draft.

---

> > ### Author Rebuttal · Reviewer_w4AT · 2026-04-04
> >
> > Thanks for your rebuttal. It has addressed my core concerns. I will increase my score to 5.

---

> > > ### Author Response · Authors · 2026-04-04
> > >
> > > Thank you for the positive feedback. We are glad that our rebuttal addressed your concerns, and we sincerely appreciate your time and consideration.

---

### Decision · Program_Chairs · 2026-04-30

**Decision:**

Accept (regular)

**Comment:**

Meta-Review Decision: Accept

Thank you to the reviewers for their constructive and consistently positive evaluations. The authors have adequately addressed the raised concerns in their rebuttal, and the paper's novelty, technical soundness, and empirical validation are well recognized. After careful consideration, I recommend Accept. This work represents a solid contribution that aligns with the conference's standards, and I look forward to seeing it presented.